# RAG-SR: Retrieval-Augmented Generation for Neural Symbolic Regression

**Hengzhe Zhang**[1,3]**, Qi Chen**[1,3]**, Bing Xue**[1,3]**, Wolfgang Banzhaf**[2]**, Mengjie Zhang**[1,3*]

[1]School of Engineering and Computer Science, Victoria University of Wellington, New Zealand
[2]Department of Computer Science and Engineering, Michigan State University, USA
[3]Centre for Data Science and Artificial Intelligence, Victoria University of Wellington, New Zealand

## Abstract

Symbolic regression is a key task in machine learning, aiming to discover mathematical expressions that best describe a dataset. While deep learning has increased interest in using neural networks for symbolic regression, many existing approaches rely on pre-trained models. These models require significant computational resources and struggle with regression tasks involving unseen functions and variables. A pre-training-free paradigm is needed to better integrate with search-based symbolic regression algorithms. To address these limitations, we propose a novel framework for symbolic regression that integrates evolutionary feature construction with a neural network, without the need for pre-training. Our approach adaptively generates symbolic trees that align with the desired semantics in real-time using a language model trained via online supervised learning, providing effective building blocks for feature construction. To mitigate hallucinations from the language model, we design a retrieval-augmented generation mechanism that explicitly leverages searched symbolic expressions. Additionally, we introduce a scale-invariant data augmentation technique that further improves the robustness and generalization of the model. Experimental results demonstrate that our framework achieves state-of-the-art accuracy across 25 regression algorithms and 120 regression tasks [1].

## 1 Introduction

Symbolic regression (SR) is a machine learning technique that searches the space of symbolic expressions to identify models that best fit a dataset (Sun et al., 2023; Fong et al., 2023). Unlike traditional regression methods, which assume a fixed model structure, SR automatically determines both the structure and parameters of the model. This flexibility allows SR to achieve both high accuracy and interpretability, making it especially valuable in fields such as physics (Udrescu et al., 2020), biology (Brunton et al., 2016), and finance (Liu & Guo, 2023), where uncovering transparent, understandable models is crucial for scientific discovery and informed decision-making.

In this paper, we focus on an automated feature construction approach to SR (Cava et al., 2019). The key idea is to generate a set of symbolic trees/features, $\Phi = \{\phi_1, \ldots, \phi_m\}$, from a dataset $(X, Y)$ to enhance the performance of an interpretable model $\mathcal{M}$, such as linear regression (Cava et al., 2019; Zhang et al., 2023a). The objective is to minimize the loss function $\mathcal{L}(\Phi; X, Y)$, defined as:

$$\mathcal{L}(\Phi; X, Y) = \frac{1}{N} \sum_{i=1}^{N} \ell\left(\mathcal{M}\left(\phi_1(X_i), \ldots, \phi_m(X_i)\right), Y_i\right), \tag{1}$$

where $N$ represents the number of instances, and $X_i$ and $Y_i$ represent the features and label for the $i$-th instance in the training data. By decomposing the SR task into the discovery of feature sets, this approach reduces the complexity of the problem. Even if each feature $\phi$ is weakly correlated with the target $Y$, the model can still perform well as long as the features collectively complement each other in predicting the target. This symbolic regression paradigm is particularly effective for

---

*Emails: {hengzhe.zhang, qi.chen, bing.xue, mengjie.zhang}@ecs.vuw.ac.nz, banzhafw@msu.edu.

[1]Source Code: https://github.com/hengzhe-zhang/RAG-SR

complex real-world problems, where the complexity of the underlying system cannot be captured by a simple equation.

Traditional SR methods, predominantly based on genetic programming (GP) (Banzhaf et al., 1998), perform gradient-free searches within the symbolic space (Jiang & Xue, 2024). While effective at exploration, these methods often lack search effectiveness due to limited guidance from accumulated knowledge during the evolutionary process. Recent advances in deep learning for SR (Biggio et al., 2021; Kamienny et al., 2022) aim to address these inefficiencies by leveraging knowledge more effectively.

Deep learning-based SR typically follows three primary paradigms: pre-trained language models (Biggio et al., 2021; Kamienny et al., 2022), reinforcement learning (Landajuela et al., 2021), and sparse supervised learning (Sahoo et al., 2018). Sparse supervised learning does not generate symbolic models directly; instead, it relies on heuristic pruning and neural architecture search to sparsify the network so that it can be converted into symbolic expressions (Li et al., 2024). In contrast, pre-trained language models and reinforcement learning can generate symbolic models directly. However, pre-trained language models require prior assumptions about the problem space, limiting their generalizability to novel tasks involving unseen functions and features. Additionally, identifying an optimal set of features for modeling complex real-world systems is time-consuming, making it impractical to generate many pairs of symbolic models and their outputs for pre-training. While reinforcement learning with a language model offers task adaptability (Landajuela et al., 2022), its low sample efficiency remains a significant drawback. Therefore, it is desirable to explore supervised learning methods that do not rely on pre-training for SR to overcome these challenges.

To develop an effective and efficient neural network for SR, we propose a novel neural network-based symbolic regression framework inspired by geometric semantic genetic programming (GSGP) (Moraglio et al., 2012). As illustrated in Figure 1, the core idea is to use a neural network to dynamically predict the best feature $\phi$ to replace an existing feature in the current set $\Phi = \{\phi_1, \ldots, \phi_m\}$, with the goal of filling the gap in the residual $R$, referred to as the desired semantics in this paper. Throughout the evolutionary process, the relationship between the semantics/outputs of each symbolic tree $\phi(X)$ and its symbolic representation $\phi$ is captured and stored in a neural semantic library, which is continuously updated in an online fashion.

One challenge with neural semantic libraries is that language models may generate features $\phi$ that are grammatically correct but irrelevant to the desired semantics $R$. In the language model domain, this is known as hallucination (Sun et al., 2024). To mitigate this, we propose a retrieval-augmented generation technique to reduce hallucination and generate symbolic trees that better align with the desired semantics. In summary, the key contributions of this paper are as follows:

- We propose a semantic descent algorithm to optimize symbolic models using a neural network with online supervised learning. The neural network continuously learns to generate symbolic trees that precisely capture the desired semantics, pushing the boundaries of deep symbolic regression to handle complex problems.
- To reduce hallucination in language models, we develop a retrieval-augmented generation mechanism. This technique makes the generated symbolic models are not only grammatically correct but also better aligned with the desired semantics, resulting in more accurate predictions.
- To better capture the relationship between desired semantics and retrieved symbolic expressions, we propose a masked contrastive loss, which more accurately generates symbolic trees by aligning the embeddings of desired semantics with those of the retrieved expressions.
- We propose a data augmentation and double query strategy to fully exploit the scale-invariant properties of feature construction-based symbolic regression, further improving the effectiveness of generated symbolic expressions.

## 2 RELATED WORK

In the domain of neural symbolic regression, a key advantage of pre-trained models is that, once pre-trained (Biggio et al., 2021; Kamienny et al., 2022), models can be reused for similar tasks

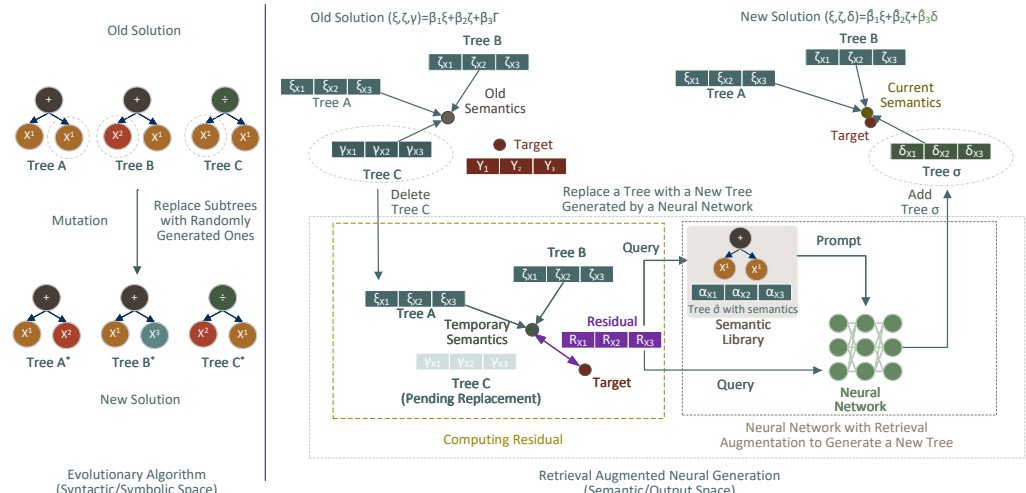

Figure 1: Comparison of the evolutionary algorithm and retrieval-augmented neural semantic library for feature construction-based symbolic regression.

without further optimization. These models are designed to solve a distribution of tasks through mechanisms such as invariance encoding (Holt et al., 2023), contrastive learning (Li et al., 2022), or conditional constraints (Bendinelli et al., 2023) to capture relationships among different SR tasks within a problem space. However, these methods may struggle with tasks beyond the scope of the pre-training data, particularly when encountering different function sets or more variables than those seen during training (Shojaee et al., 2024a; Meidani et al., 2024). Fine-tuning could alleviate the misalignment between training and target tasks, through approaches like reinforcement learning (Holt et al., 2023) or using imitation learning to learn successful mutations (Kamienny et al., 2023). However, fine-tuning large pre-trained language models can be challenging. Thus, exploring how online learning techniques can be applied exclusively to enhance SR remains a promising and underexplored direction.

Reinforcement learning (RL), on the other hand, learns the probability distribution of promising symbolic models (Landajuela et al., 2021; Xu et al., 2024) by interacting with the environment, allowing it to adapt to different function sets for various tasks. However, deep symbolic optimization via RL often suffers from low sample efficiency, requiring integration with GP (Mundhenk et al., 2021) or Monte Carlo Tree Search (MCTS) (Xu et al., 2024) techniques to improve performance. Furthermore, RL typically simplifies feedback to a scalar reward, such as mean squared error (Landajuela et al., 2021), which limits the richness of information provided during the search process. A more effective approach would involve using a loss vector rather than a scalar loss to provide richer feedback and enhance overall search effectiveness.

Sparse supervised learning methods, such as deep equation learners (Sahoo et al., 2018) and efficient symbolic policy learning (Guo et al., 2024), aim to derive interpretable symbolic models by regularizing neural networks (Zhang et al., 2023c). However, since the $L_0$ norm is non-differentiable, these techniques often rely on heuristic pruning approaches to convert neural networks into interpretable expressions. Additionally, they typically require neural architecture search methods to identify suitable architectures before gradient-based training (Li et al., 2024).

Evolutionary symbolic regression is primarily based on the GP framework, which automatically discovers symbolic models without predefined structures to fit the training data (Fong et al., 2023). Recently, semantic GP has gained substantial attention (Moraglio et al., 2012; Zhang et al., 2023b). Unlike traditional GP, which operates in the syntactic/symbolic space, semantic GP works in the semantic/output space. By focusing on semantic space, solution generation operators can ensure that the newly generated solutions have more predictable behavior, such as guaranteed loss reduction—something conventional GP operators often lack. A key challenge in semantic GP is generating GP trees that satisfy the desired semantics (Moraglio et al., 2012). A common strategy is to build a semantic library that stores evaluated GP trees (Pawlak et al., 2014). In semantic mutation, this library is searched for trees that closely match the target semantics, and the best-matching tree

---

**Algorithm 1** Semantic Descent

---

1: **Input:** Features $\Phi = \{\phi_1, \ldots, \phi_m\}$, semantics library $\mathcal{L}$, neural network model $\mathcal{N}$, current semantics $\Phi(X)$, target $Y$, neural generation probability $P_{\text{neural}}$
2: **Output:** Updated features $\Phi$
3: $\mathcal{O} \leftarrow$ Random permutation of $\{1, 2, \ldots, m\}$ $\qquad\qquad\qquad\qquad\qquad$ ▷ Shuffle tree indices
4: **for** each $i \in \mathcal{O}$ **do**
5: $\qquad \tilde{\phi}_i(X) \leftarrow \frac{\phi_i(X) - \mu_i}{\sigma_i}$ $\qquad\qquad\qquad\qquad\qquad\qquad\qquad\qquad$ ▷ Normalized feature
6: $\qquad \Phi(X)^{\text{temp}} \leftarrow \Phi(X) - \beta_i \tilde{\phi}_i(X)$
7: $\qquad \mathbf{R} \leftarrow Y - \Phi(X)^{\text{temp}}$ $\qquad\qquad\qquad\qquad\qquad\qquad\qquad$ ▷ Compute residual $R$
8: $\qquad$ **if** rand() $< P_{\text{neural}}$ **then**
9: $\qquad\qquad \phi_i \leftarrow \mathcal{N}(\mathbf{R}, \mathcal{L})$ $\qquad\qquad\qquad\qquad$ ▷ Generate new tree using neural model
10: $\qquad\qquad \Phi(X) \leftarrow \Phi(X)^{\text{temp}}$
11: $\qquad\qquad$ **continue** $\qquad\qquad\qquad\qquad\qquad\qquad\qquad$ ▷ Proceed to next tree
12: $\qquad$ **end if**
13: $\qquad \phi_{\text{new}} \leftarrow \texttt{ExactRetrieval}(\mathbf{R}, \phi_i, \mathcal{L})$
14: $\qquad \phi_i, \Phi(X) \leftarrow \texttt{ExactReplacement}(\phi_{\text{new}}, \phi_{\text{new}}(X), \Phi(X), \Phi(X)^{\text{temp}}, \mathbf{R}, Y)$
15: **end for**

---

is selected. However, this approach relies solely on existing building blocks without leveraging historical knowledge to create new ones. To address this issue, it is crucial to incorporate deep learning techniques to learn from the evolutionary learning process and generate better symbolic models that align with the desired semantics.

## 3 ALGORITHM

The proposed method is based on an evolutionary algorithm framework, encompassing solution initialization, generation, evaluation, selection, and archive maintenance. This work primarily focuses on the solution generation phase, introducing a neural semantic library for solution generation, designed to explicitly retain and apply knowledge throughout the evolutionary process. Solution generation consists of two primary components: semantic descent and retrieval-augmented generation.

### 3.1 SEMANTIC DESCENT

In this work, we propose Semantic Descent (SD), an iterative optimization procedure designed to improve model performance by selectively replacing suboptimal features. Unlike methods such as geometric semantic GP (Moraglio et al., 2012) or gradient boosting (Feng et al., 2018), which incrementally add new features to minimize error, SD focuses on replacing existing trees in the model with more informative ones. This approach helps maintain a compact model structure while continuously improving accuracy.

At each iteration, a tree $\phi_i$ is randomly selected from the set of trees $\{\phi_1, \ldots, \phi_m\}$ that define the semantics/outputs of the model $\Phi(X) = \beta_1 \phi_1(X) + \cdots + \beta_m \phi_m(X) + \alpha$, where $\beta$ represents the coefficients and $\alpha$ is the intercept. The contribution of $\phi_i$ is temporarily removed, resulting in temporary semantics $\Phi^{\text{temp}}(X) = \Phi(X) - \beta_i \phi_i(X)$. The residual $R = Y - \Phi^{\text{temp}}(X)$ of the model is then computed, where $Y$ is the target output. The residual $R$ represents the difference between the prediction and the target.

As shown in Algorithm 1, the core idea of SD is to fill the gap in the residual $R$ by replacing the current tree $\phi_i$ with a better alternative, either generated by a neural model $\mathcal{N}$ (line 9) or retrieved from a semantic library $\mathcal{L}$ (line 13). The semantic library $\mathcal{L}$ stores all previously evaluated symbolic trees and subtrees $\psi$ along with their semantics/outputs $\psi(X)$. The neural model $\mathcal{N}$ learns the mapping between the semantics $\psi(X)$ and the corresponding symbolic tree $\psi$. This enables the neural network $\mathcal{N}$ to construct a new feature $\phi_{\text{new}}$ using $R$ as input, thereby generating a new feature to reduce the model's error.

The probability of generating new trees using the neural network is $P_{\text{neural}}$, detailed in Section 3.2. The probability of retrieving a tree from the semantic library is $1 - P_{\text{neural}}$. The key idea of exact retrieval is to search the library for the tree that most closely matches the desired semantics, i.e., the residual $R$, as detailed in Appendix C. Since the linear regression model automatically adjusts

feature magnitudes and intercepts, the residual $R$ is normalized using the $L_2$ norm before being used as input for retrieval or neural generation, i.e., $R \leftarrow \frac{R - \bar{R}}{||R||_2}$. The replacement process is repeated iteratively until all trees $\phi$ within the solution $\Phi$ have been traversed. By focusing on feature replacement instead of addition, SD enables efficient model refinement while maintaining a fixed feature set, allowing for both interpretability and performance improvements.

## 3.2 RETRIEVAL-AUGMENTED GENERATION

To learn the mapping between symbolic trees, $\phi$, and their corresponding semantics, $\phi(X)$, the process involves three steps: First, the trees and semantics are collected from the evolutionary process (Section 3.2.2). Next, they are converted into training data using specially designed encoding rules (Section 3.2.1). Finally, a neural network is trained on the collected data (Section 3.2.3), using cross-entropy loss and masked contrastive loss (Section 3.2.4).

### 3.2.1 DATA COLLECTION AND NETWORK TRAINING

The semantic library $\mathcal{L}$ is dynamically constructed during the evolutionary process. During solution evaluation, each subtree $\psi$ and its corresponding semantics $\psi(X)$ are stored in a first-in-first-out queue $Q$ with an upper limit of 10,000 entries for training the neural network and future retrieval. To facilitate efficient retrieval, a k-dimensional tree (k-d tree) is constructed using the semantics stored in $Q$ at the end of each generation in the evolutionary process, reducing query complexity to $O(\log(N))$, where $N$ is the number of stored trees. The neural network is also trained at the end of each generation. To prevent unnecessary training, an internal validation set monitors performance degradation. If the validation loss does not increase, network training is skipped for that generation to save computational resources. Nevertheless, the retrieval library is updated even when network training is bypassed, ensuring that knowledge base is continuously updated throughout the evolutionary process.

### 3.2.2 ENCODING AND DECODING RULES FOR SYMBOLIC TREES

To generate valid expressions without requiring an explicit end token, we designed a specialized encoding and decoding scheme. Symbolic trees are encoded using a level-order traversal method, specifically breadth-first search, to convert the tree into a linear sequence. To maintain interpretability, the number of functions in the symbolic tree is capped at $n_F$. Given this limit and the maximum number of children any function can have, $\alpha_{\max}$, the maximum number of terminal nodes is:

$$n_T = 1 + (n_F \times (\alpha_{\max} - 1)). \tag{2}$$

Given $n_F$ and $n_T$, the neural network outputs a fixed-length sequence with the first $n_F$ tokens decoded as functions or terminals,while the remaining $n_T$ tokens are restricted to terminals by setting function probabilities to zero during decoding. This formulation reframes the symbolic tree generation task as a multi-class classification problem. Detailed pseudocode for the encoding and decoding processes is provided in Appendix E.

### 3.2.3 OVERALL ARCHITECTURE FOR RETRIEVAL-AUGMENTED GENERATION

As shown in Figure 2, the neural architecture consists of two main components: a Multilayer Perceptron (MLP) and a Transformer model. Their relationship is defined as:

$$\mathbf{O} \in \mathbb{R}^{B \times L \times S} = \text{Transformer Decoder} \left( \text{Transformer Encoder}(\hat{\phi}) \oplus \text{MLP}(R) \right) \cdot \mathbf{W}^T. \tag{3}$$

Each of the two components plays a distinct and complementary role in generating a symbolic tree $\phi$ based on the desired semantics $R$. The MLP transforms raw semantics into a meaningful feature representation that can guide the generation of the symbolic tree. On the other hand, the Transformer encoder processes the nearest symbolic tree $\hat{\phi}$, retrieved from the semantics library $\mathcal{L}$, which serves as a prompt to reduce hallucination. The outputs of the MLP and Transformer are concatenated and then passed through a Transformer decoder to generate a sequence of $L$ tokens. $\mathbf{W} \in \mathbb{R}^{S \times D}$ is a linear layer that projects the output of the Transformer decoder $\mathbf{H}_{\text{Decoder}} \in \mathbb{R}^{B \times L \times D}$ into the symbol space $\mathbf{O} \in \mathbb{R}^{B \times L \times S}$, where $S$ is the number of unique symbols. These tokens are subsequently decoded into a valid symbolic tree.

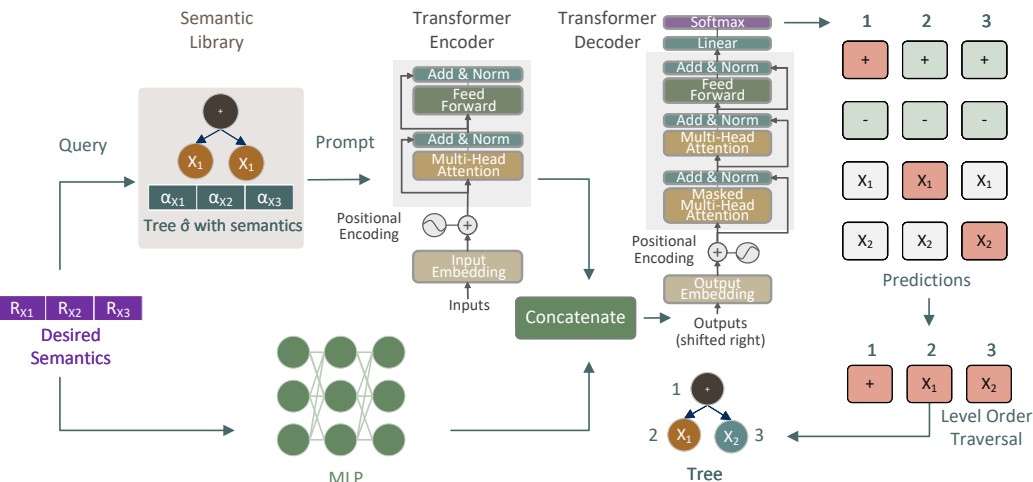

Figure 2: Neural network architecture for symbolic tree generation.

**Intention Encoding:** The desired semantics $R \in \mathbb{R}^{B \times N}$ is processed through an MLP to produce a feature matrix $\mathbf{F}_{\text{MLP}} \in \mathbb{R}^{B \times K}$, where $K$ is the dimensionality of the hidden layer. The MLP consists of $N_L$ layers, and at each layer $i$, the transformation is defined as:

$$\mathbf{x}_{i+1} = \text{Dropout}_i \left( \text{SiLU}_i \left( \text{BN}_i \left( \mathbf{W}_i \cdot \mathbf{x}_i + \mathbf{b}_i \right) \right) \right) + \mathbf{x}_i, \tag{4}$$

where $\mathbf{x}_i \in \mathbb{R}^{B \times K}$ is the input to the $i$-th layer, $\mathbf{W}_i \in \mathbb{R}^{K \times K}$ is the weight matrix, $\mathbf{b}_i \in \mathbb{R}^K$ is the bias vector, $\text{BN}_i$ denotes the batch normalization layer, $\text{SiLU}_i$ is the Sigmoid Linear Unit activation function (Elfwing et al., 2018), and $\text{Dropout}_i$ is the dropout layer with a specified dropout rate. This MLP layer results in a feature matrix $\mathbf{F}_{\text{MLP}} \in \mathbb{R}^{B \times K}$, which is then passed through a linear layer to match the dimensionality from $K$ to $D$, yielding $\mathbf{F}_{\text{MLP}}^{\text{mapped}} \in \mathbb{R}^{B \times D}$, where $D$ is the dimensionality of the Transformer-encoded representation.

**Retrieval-Augmented Encoding:** For the desired semantics $R$, a KD-Tree is used to retrieve the nearest symbolic tree $\hat{\phi}$ from the semantic library $\mathcal{L}$, based on Euclidean distance and subject to the constraint that the tree contains no more than $n_F$ nodes. The retrieved tree $\hat{\phi}$ is then processed through an embedding layer to generate $\mathbf{V}_{\hat{\phi}} \in \mathbb{R}^{B \times L \times E}$, where $L$ is the sequence length of the tree encoding and $E$ is the dimensionality of the embedding space. The embedding layer consists of an embedding matrix $\mathbf{E} \in \mathbb{R}^{S \times E}$. The embedded representation $\mathbf{V}_{\hat{\phi}}$ is then encoded using the Transformer model to produce a symbolic model embedding $\mathbf{H}_{\text{Transformer}} \in \mathbb{R}^{B \times L \times D}$. The Transformer encoder applies self-attention and feedforward layers with residual connections as follows:

$$\begin{aligned}
\mathbf{H}_{\text{Self-Attn}} &= \text{LayerNorm} \left( \mathbf{V}_{\hat{\phi}} + \text{SelfAttention} \left( \mathbf{V}_{\hat{\phi}} \right) \right) \quad \in \mathbb{R}^{B \times L \times K}, \\
\mathbf{H}_{\text{Transformer}} &= \text{LayerNorm} \left( \mathbf{H}_{\text{Self-Attn}} + \text{FeedForward} \left( \mathbf{H}_{\text{Self-Attn}} \right) \right) \quad \in \mathbb{R}^{B \times L \times D}.
\end{aligned} \tag{5}$$

**Decoding:** The combined feature representation $\mathbf{H}_{\text{Combined}} = \mathbf{F}_{\text{MLP}}^{\text{mapped}} \oplus \mathbf{H}_{\text{Transformer}} \in \mathbb{R}^{B \times (L+1) \times D}$ is fed into a Transformer decoder to generate the contextual embeddings $\mathbf{H}_{\text{Decoder}} \in \mathbb{R}^{B \times L \times D}$. The decoding process is performed auto-regressively, utilizing a greedy decoding strategy.

### 3.2.4 LOSS FUNCTION

**Masked Contrastive Loss:** The intention encoding should ideally learn useful knowledge not only from target expressions $\phi$ but also from the retrieved symbolic expressions $\hat{\phi}$. In parallel, the retrieval-augmented encoding should be aware of the semantics of the nearest symbolic expressions. To fulfill these objectives, we propose a contrastive loss that aligns the embeddings from both the intention encoding and the retrieval-augmented encoding components.

Given the nearest semantics $\hat{\phi}(X) \in \mathbb{R}^{B \times N}$, it is processed through a MLP to generate a feature matrix of nearest semantics $\mathbf{F}_{\text{nearest}} \in \mathbb{R}^{B \times K}$. Simultaneously, the embedding of the symbolic model

$\mathbf{H}_{\text{Transformer}} \in \mathbb{R}^{B \times L \times D}$ is averaged along the sequence length dimension to produce the averaged embedding $\mathbf{H}_{\text{avg}} \in \mathbb{R}^{B \times D}$. Then, the InfoNCE loss (Oord et al., 2018), a popular objective in contrastive learning, is employed to maximize the similarity between the nearest semantics feature matrix $\mathbf{F}_{\text{nearest}}$ and the averaged symbolic embeddings $\mathbf{H}_{\text{avg}}$, while minimizing similarity with negative samples from the same batch. To alleviate false negatives, i.e., when two samples in a batch are semantically similar, the InfoNCE loss is masked by a mask matrix $mask$. The mask matrix is designed such that non-diagonal elements with an absolute cosine similarity greater than 0.99 are marked as false (indicating false negatives), while all other entries are marked as true. The masked InfoNCE loss is formally defined as:

$$\mathcal{L}_{\text{InfoNCE}} = -\frac{1}{B} \sum_{i=1}^{B} \log \frac{\exp\left(\text{sim}(\mathbf{F}_{\text{nearest}}[i], \mathbf{H}_{\text{avg}}[i])/\tau\right)}{\sum_{j=1}^{B} \exp\left(\text{sim}(\mathbf{F}_{\text{nearest}}[i], \mathbf{H}_{\text{avg}}[j]) \cdot \text{mask}/\tau\right)}, \quad (6)$$

where $\text{sim}(\cdot, \cdot)$ denotes cosine similarity, and $\tau$ is a temperature parameter controlling the sharpness of the softmax function. This contrastive loss ensures that the nearest semantics are closely aligned with their corresponding symbolic representations in the embedding space, while differentiating them from unrelated samples.

**Cross-Entropy Loss:** The model is also trained using cross-entropy loss over the sequence of $L$ symbols. Let $\mathbf{o}_{\text{true}}^i \in \mathbb{R}^S$ denote the one-hot encoded ground truth for the $i$-th position, and $\mathbf{o}_{\text{pred}}^i \in \mathbb{R}^S$ denote the predicted probability distribution at that position. Formally, the cross-entropy loss for each sequence is defined as $\mathcal{L}_{\text{cross-entropy}} = -\sum_{i=1}^{L} \mathbf{o}_{\text{true}}^i \cdot \log(\mathbf{o}_{\text{pred}}^i)$. The final loss $\mathcal{L}$ is a weighted sum of the cross-entropy loss and the contrastive loss:

$$\mathcal{L} = \mathcal{L}_{\text{cross-entropy}} + \lambda \cdot \mathcal{L}_{\text{InfoNCE}}, \quad (7)$$

where $\lambda$ is a hyperparameter that balances the contributions of the two losses.

## 3.3 Data Augmentation and Double Query

In linear regression, the sign of coefficients is automatically adjusted, so the sign of the semantics is not crucial. However, the training data may only include one side of a training pair $(\psi, \psi(X))$, without considering its opposite, $(\psi, -\psi(X))$. Consequently, when the desired semantics is $-\psi(X)$, the model may fail to generate the correct symbolic tree $\psi$. To address this issue, we augment the training data by including both $(\psi, \psi(X))$ and $(\psi, -\psi(X))$ pairs:

$$\mathcal{T} \leftarrow \mathcal{T} \cup \{(\psi, -\psi(X)) \mid (\psi, \psi(X)) \in \mathcal{T}\}. \quad (8)$$

During decoding, both $R$ and $-R$ are used to query the neural network, generating candidate trees $\phi$ and $\phi'$. The tree with the highest probability is selected as the final symbolic model. This technique, referred to as double query (DQ), allows the model to generate symbolic trees with sign-insensitive semantics, thereby improving the effectiveness of neural generation.

## 4 Experiments

This section is divided into two parts. The first part evaluates the effectiveness of the proposed components in improving the prediction accuracy of the neural semantic library. The second part investigates the performance of integrating the SR method with the retrieval-augmented neural semantic library. It compares this integrated approach to state-of-the-art SR methods.

## 4.1 Experimental Results of Neural Semantic Library

**Experimental Settings:** To evaluate the effectiveness of the proposed techniques in enhancing the learning capabilities of the neural semantic library, we conduct the first experiment on synthetic data. The objective is to evaluate how various components contribute to the learning effectiveness of the neural semantic library. In this experiment, 10 variables and 50 training instances are randomly drawn from a Gaussian distribution $\mathcal{N}(0, 100)$. Then, a total of 10000 symbolic expressions with random heights $h \in [0, 5]$ are generated using the grow method (Banzhaf et al., 1998) from GP and evaluated on the randomly generated data. The maximum number of functions $n_F$ is set to 5, and

expressions exceeding this limit are filtered out. To avoid redundancy, only one semantically equivalent GP tree is retained, ensuring no symbolic expressions overlap between training and test sets. This setup ensures that the final metric reflects the ability of the neural network to learn patterns and generalize to unseen data, rather than simply fitting to previously seen examples. A total of 80% of the symbolic models are used for training, while the remaining 20% are reserved for testing. The evaluation metric is the edit distance (Matsubara et al., 2022; Bertschinger et al., 2023) between the generated symbolic tree and the ground truth, where a smaller distance indicates that the neural semantic library generates more effective building blocks, significantly aiding the evolutionary algorithm in finding optimal solutions. Each experiment is run 5 times to ensure stable and reliable results.

**Parameter Settings:** For the neural network, the dropout rate is set to 0.1. The MLP consists of 3 layers, while both the encoder and decoder Transformers have 1 layer each. The hidden layer size is set to 64 neurons. A learning rate of 0.01 and a batch size of 64 are used. Early stopping with a patience of 5 epochs is employed to prevent overfitting. The weight of contrastive loss $\lambda$ is set to 0.05.

**Experimental Results (Edit Distance):** The experimental results for edit distance on the test set are presented in Figure 3. First, comparing neural generation with simple retrieval from the library (W/O NN), neural generation performs better by a large margin, indicating the effectiveness of using a neural network for symbolic tree generation. As for the ablation results of components, the results show that including all components achieves the lowest median edit distance, indicating that the combination of all proposed techniques provides the best overall performance. Among the components, the RAG technique has the most significant impact, highlighting that external knowledge from the semantic library significantly improves the neural network's ability to generate relevant symbolic trees. Data augmentation (DA) also plays a crucial role, ranking as the second most important component. Without DA, the model struggles to handle the scale-invariant nature of feature construction, leading to worse performance. The compact boxplots reflect the consistency and reliability of these components. Dropout has a moderate positive effect, indicating that overfitting control techniques are helpful for training the neural semantic library. Similarly, contrastive learning (CL) shows a moderate impact, confirming the effectiveness of using contrastive loss to align the intention encoding with retrieval augmentation encoding components. Finally, DQ also improves effectiveness, showing that even simply generating multiple solutions during inference can lead to better solutions, which aligns with findings from large language models (Wang et al., 2023). The impact of DQ becomes more pronounced in the absence of DA, suggesting that DA partially compensates for the lack of DQ.

**Experimental Results (Running Time):** The running time comparisons in Figure 4 demonstrate that RAG moderately increases the overall running time. However, one advantage of incorporating RAG into the component is that new trees can be seamlessly added to the retrieval library to improve accuracy without requiring model fine-tuning, making the algorithm efficient for application in an online learning setting. For DA and DQ, removing these components reduces the running time from 44 seconds to 35 and 29 seconds, respectively, indicating that they do introduce some computational overhead. However, given the accuracy improvements they provide, the increase in computational time is acceptable. Although removing both DA and DQ significantly reduces computational cost, the substantial loss of edit distance from 3.82 to 4.35 outweighs the benefit of faster execution.

**Examples of Generated Trees:** Table 1 provides examples of symbolic trees generated by the neural network with and without retrieval augmentation, along with the retrieved trees. The results demonstrate that the retrieved trees share certain similarities with the ground truth, such as variable usage. These results validate that providing the retrieval tree as a prompt helps the neural network generate more relevant trees, reducing hallucination compared to relying solely on the desired semantics.

## 4.2 EXPERIMENTS OF RAG-SR

**Datasets:** In this study, we primarily focus on 120 black-box datasets from the PMLB benchmark (Olson et al., 2017), which are particularly challenging for pre-training methods (Kamienny et al., 2022) due to the potential absence of simple symbolic expressions to model these datasets. The results on the 119 Feynman and 14 Strogatz datasets are presented in Appendix L.2.

Table 1: Examples of symbolic trees generated by the retrieval-augmented neural network, simple neural network, retrieval library, and ground truth.

| RAG-NN Generated Tree (Distance) | Simple NN Generated Tree (Distance) |
|---|---|
| Sin(Sin(ARG3)) (0) | Cos(Cos(Cos(ARG9))) (4) |
| AQ(ARG7, ARG8) (0) | Log(Max(ARG7, ARG7)) (3) |
| Max(ARG1, ARG8) (0) | Subtract(ARG1, ARG1) (2) |
| Sqrt(Sqrt(ARG2)) (0) | Log(Log(ARG2)) (2) |
| Subtract(ARG6, ARG7) (0) | Max(ARG7, ARG7) (2) |

| Retrieval Tree (Distance) | Ground Truth Tree |
|---|---|
| Sin(ARG3) (1) | Sin(Sin(ARG3)) |
| Log(Neg(Max(AQ(ARG8, ARG0), AQ(ARG7, ARG8)))) (6) | AQ(ARG7, ARG8) |
| Max(add(Abs(Sin(Cos(ARG6))), ARG8), ARG1) (6) | Max(ARG1, ARG8) |
| Square(Log(ARG2)) (2) | Sqrt(Sqrt(ARG2)) |
| Subtract(ARG7, ARG6) (2) | Subtract(ARG6, ARG7) |

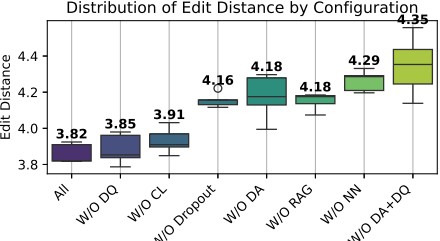

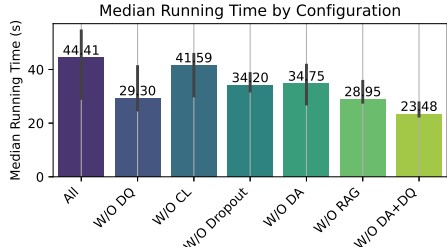

Figure 3: Ablation study of components based on edit distance on the test set.

Figure 4: Ablation study of components with respect to running time (training and inference).

**Evaluation Protocol:** The evaluation follows the established procedures of state-of-the-art symbolic regression benchmarks (La Cava et al., 2021). Specifically, each dataset is split into training and testing sets with a 75:25 ratio, and experiments are repeated 10 times for robustness. The $R^2$ score on the test set is used as the evaluation metric. To better handle categorical variables, we use a target encoder (Micci-Barreca, 2001). Furthermore, to prevent any single feature from disproportionately influencing the semantics, all input features are normalized using min-max scaling (Raymond et al., 2020).

**Parameter Settings:** For GP, we follow conventional parameter settings: a population size of 200 and a maximum of 100 generations. Each solution consists of 10 trees, representing 10 features. The probability of using neural generation, $P_{\text{neural}}$, is set to 0.1.

**Experimental Results (Accuracy):** The experimental results on SRBench are presented in Figure 5. The proposed method, RAG-SR, outperforms all state-of-the-art symbolic regression and machine learning techniques in terms of $R^2$ scores. Notably, it surpasses the TPSR method (Shojaee et al., 2024a), which combines MCTS with a pre-trained end-to-end Transformer (Kamienny et al., 2022). The improvement is statistically significant, as confirmed by the Wilcoxon signed-rank test with Benjamini-Hochberg correction, shown in Figure 6. This indicates the effectiveness of using a purely online training language model for learning symbolic expressions. Compared to SBP-GP (Pawlak et al., 2014), which is a purely retrieval-based geometric semantic GP that does not use a neural network, the significant advantage of RAG-SR demonstrates the effectiveness of using a neural network to dynamically generate symbolic models.

**Experimental Results (Complexity):** The model complexity of RAG-SR follows the definition of SRBench, where the final model is converted into a SymPy-compatible expression, and the number of nodes in the symbolic tree is counted as a measure of complexity. As shown in Figure 5, RAG-SR produces models that are an order of magnitude smaller in size compared to PS-Tree (Zhang et al., 2022), which is a piecewise SR method that ranks second in $R^2$ scores in Figure 5. The Pareto front of test $R^2$ scores and model size rank is shown in Figure 7, where RAG-SR appears on the first Pareto front, indicating that RAG-SR achieves a good balance between accuracy and model complexity.

**Experimental Results (Training Time):** The training time of RAG-SR is comparable to that of FEAT, a standard feature-construction-based SR method (Cava et al., 2019), suggesting that the

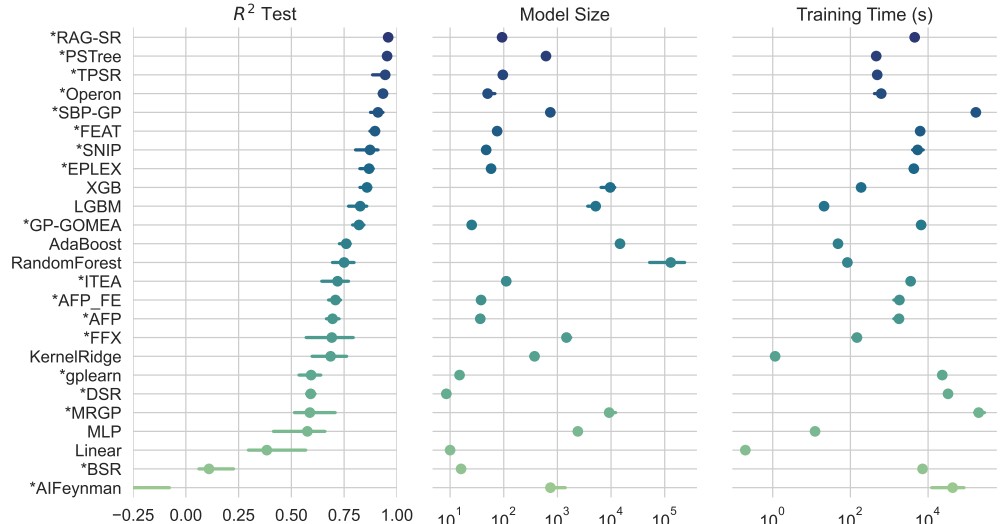

Figure 5: $R^2$ scores, model sizes, and training time of 25 algorithms on 120 regression problems.

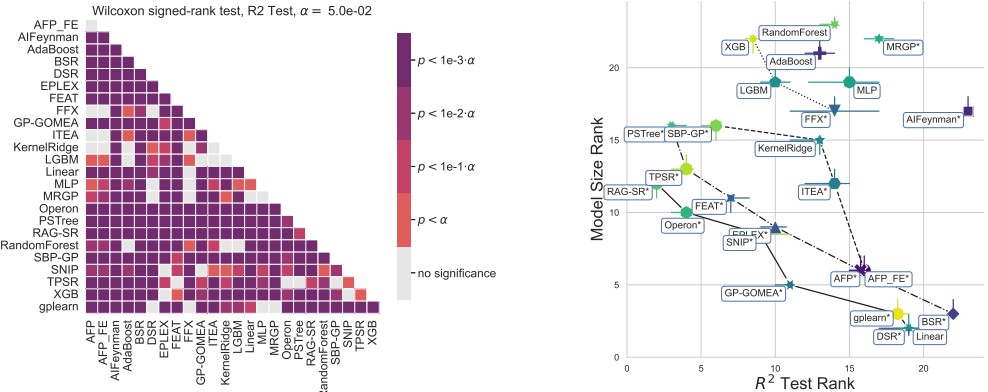

Figure 6: Pairwise statistical comparisons of test $R^2$ scores on regression problems.

Figure 7: Pareto front of the rank of test $R^2$ scores and model size for different algorithms.

computational cost of learning a neural semantic library is within an acceptable range. However, compared to TPSR, which directly leverages a pre-trained model to guide SR without requiring fine-tuning, RAG-SR is an order of magnitude slower. This discrepancy is partly due to the fact that, in the current implementation, all neural networks in RAG-SR are trained on a CPU due to limited computational resources. Training the neural networks on a GPU could potentially reduce the computational time of RAG-SR.

## 5 Conclusions

In this paper, we propose a novel feature construction-based SR method with a retrieval-augmented neural semantic library. Ablation studies confirm that the retrieval augmentation mechanism effectively mitigates the issue of hallucination, enabling the generation of more accurate symbolic trees that align with the desired symbolic trees. Furthermore, data augmentation and double query techniques effectively improve the neural network's ability to generate symbolic trees that account for the scale-invariant characteristics of feature construction-based SR. Experimental results on large-scale symbolic regression benchmarks demonstrate that RAG-SR significantly outperforms state-of-the-art SR techniques, including those guided by pre-trained language models. For future directions, introducing constraints on model complexity may help reduce the risk of overfitting, particularly with datasets that contain noise or limited samples, presenting a promising direction for future research.

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

## A  ALGORITHM WORKFLOW

The algorithm workflow, shown in Figure 8, includes the following steps:

- **Solution Initialization:** The population is initialized using random symbolic trees generated through ramped half-and-half methods (Banzhaf et al., 1998). In this approach, half of the GP trees are initialized with random depths, while the other half are initialized at the maximum initial depth. Each solution contains $m$ symbolic trees, with each tree representing a unique feature.

- **Solution Evaluation:** Features $\{\phi_1(X), \ldots, \phi_m(X)\}$ are constructed from the symbolic trees $\Phi = \{\phi_1, \ldots, \phi_m\}$ and the training data $X$. These features are evaluated via ridge regression using efficient leave-one-out cross-validation to compute the loss $\mathcal{L}(\Phi(X))$.

- **Semantic Library Construction:** To reduce the dimensionality of inputs to the neural network and optimize memory usage, a subset of hard instances is selected every 10 generations. Specifically, all training instances $X_i \in X$ are ranked by the median loss across all solutions, i.e., $\text{Median}(\mathcal{L}(\Phi(X_i)) \mid \Phi \in P)$, and the top $k = 50$ instances, ranked in descending order, are chosen. The instances are further sorted by their corresponding training labels $Y_i$. Consequently, the positions of some hard instances remain unchanged after instance selection, which may facilitate the continuous training of the neural network. The semantics of these selected instances are stored in a retrieval library, which is used for training the neural semantic library and for exact retrieval. It is important to note that instance selection is solely for semantic library construction and semantic descent. Solution evaluation is conducted on the entire training dataset.

  - **Retrieval Library:** Symbolic trees and their corresponding semantics are stored in a first-in-first-out queue $\mathcal{Q}$, which can hold up to 10,000 symbolic trees. A KD-Tree is used for efficient retrieval, reducing the search complexity from $O(N)$ to $O(\log(N))$. For neural network training, a subset of symbolic trees from $\mathcal{Q}$, where function nodes are fewer than or equal to five, is used to construct a separate KD-Tree for rapid retrieval of eligible trees.

  - **Neural Semantic Library:** The neural network is trained using pairs of symbolic trees and their corresponding semantics from the retrieval library. The architecture, described in Section 3.2, combines a multi-layer perceptron with a Transformer to process desired semantics with retrieved symbolic trees.

- **Solution Selection:** Lexicase selection (Helmuth et al., 2014) is used to identify promising solutions based on their objective values. Solutions are iteratively eliminated by applying randomly constructed criteria. In each round, a random instance $i \in [0, N]$ is selected, and solutions with losses greater than the median absolute deviation are eliminated (La Cava et al., 2019). This elimination process continues until only one solution remains, which is then selected as the promising solution. For a population of $P$ solutions, $P$ solutions are selected by iteratively applying these steps.

- **Solution Generation:** New solutions are generated based on the selected promising solutions using a memetic algorithm framework. First, semantic descent performs a local search to improve a solution, followed by evolutionary search to explore the search space and mitigate the risk of local optima:

  - **Semantic Descent:** As detailed in Section 3.1, the residual $R$ is used to query the neural semantic library or the retrieval library to either generate or retrieve GP trees. The neural generation process is described in Section 3.2. For retrieving GP trees, Appendix C explains the strategy to retrieve relevant GP trees from the library based on desired semantics. Since the retrieved features have already been evaluated, the loss reduction can be easily calculated after replacement, as explained in Appendix D. If replacing the tree does not reduce the loss, the current round of replacement is skipped.

  - **Evolutionary Search:** New symbolic trees are generated using genetic programming operators such as random subtree crossover and mutation (Banzhaf et al., 1998). For $m$ symbolic trees, the crossover and mutation operators are applied $m$ times with their respective probabilities to ensure sufficient exploration of the search space.

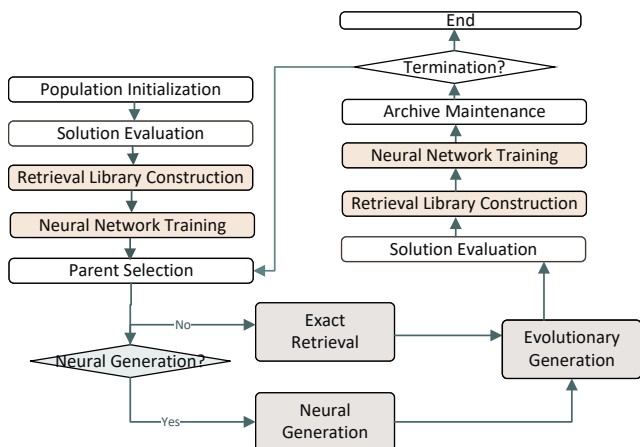

Figure 8: Workflow of neural semantic library-based symbolic regression.

- **Archive Maintenance:** The historically best-performing solution is stored in an archive for making predictions on unseen data.

The processes of solution evaluation, library construction, solution selection, solution generation and archive maintenance are repeated until the stopping criteria are met.

## B LIBRARY UPDATE

The library is updated at the end of each generation during the evolutionary process. As detailed in Algorithm 2, the library is managed using two first-in-first-out queues, $\mathcal{T}$ and $\mathcal{S}$, which store the most recent 10,000 trees and their corresponding semantics, respectively. The semantics $\psi(X)$ are first normalized to $\psi_{\text{norm}}(X)$ to ensure that retrieval is insensitive to scale. This normalization is necessary because the linear regression algorithm can automatically determine the scaling coefficient $\beta$ for each constructed feature. Subsequently, all GP trees are appended to the queue $\mathcal{T}$, while their corresponding semantics are appended to the queue $\mathcal{S}$ and stored in a hash set $\mathcal{H}_{\text{seen}}$.

To control memory usage, the hash set prevents duplicate GP trees from being added to the library. When the queues $\mathcal{T}$ and $\mathcal{S}$ reach their maximum limit $T_{\text{max}}$, the oldest entries are removed, as illustrated in Line 21.

For computational efficiency, the semantics of each subtree $\psi(X)$ are precomputed during the solution evaluation phase. Consequently, during the library update phase, it is unnecessary to recompute the semantics of each tree $\psi$ based on the data $X$. This precomputation ensures that the library update process is computationally efficient.

Additionally, for efficient querying in exact retrieval, a KD-Tree is constructed at the end of the library update. To efficiently retrieve symbolic expressions that meet the function node limits of neural generation, another KD-Tree is constructed based on symbolic expressions with a maximum of $n_F$ function nodes. More details on the construction of this KD-Tree for neural generation are provided in Appendix F. Together, the two queues, $\mathcal{T}$ and $\mathcal{S}$, along with the two KD-Trees, form the semantic library $\mathcal{L}$.

## C EXACT RETRIEVAL OF THE SYMBOLIC TREE

Exact retrieval is directly using symbolic trees from the retrieval library that align with the desired semantics $R$, complementing neural generation. This occurs with a probability of $1 - P_{\text{neural}}$, as described in Algorithm 1. In this process, the desired semantics $R$ are used to retrieve the top $\kappa = 10$ symbolic trees with the smallest distances to $R$, as shown in Algorithm 3. These $\kappa$ trees

---

**Algorithm 2** Library Update

---

1: **Input:** Population of individuals $P$, tree queue $\mathcal{T}$, semantics queue $\mathcal{S}$, hash set of seen semantics $\mathcal{H}_{\text{seen}}$, maximum tree limit $T_{\text{max}}$
2: **Output:** Updated tree queue $\mathcal{T}$ and semantics queue $\mathcal{S}$
3: **for** each individual $\Phi \in P$ **do**
4:      **for** each pair $(\psi(X), \psi)$, where $\psi$ is a GP tree/subtree in $\Phi$ and $\psi(X)$ is its corresponding semantics **do**
5:          **if** $\|\psi(X)\| = 0$ **then**
6:              **Continue**          $\triangleright$ Skip semantics with zero norm
7:          **end if**
8:          $\psi_{\text{norm}}(X) \leftarrow \frac{\psi(X)}{\|\psi(X)\|}$          $\triangleright$ Normalize semantics
9:          **if** $\psi_{\text{norm}}(X)$ contains NaN or $\infty$ **then**
10:            **Continue**          $\triangleright$ Skip invalid trees
11:          **end if**
12:          **if** $\psi_{\text{norm}}(X) \in \mathcal{H}_{\text{seen}}$ **then**
13:            **Continue**          $\triangleright$ Skip already seen semantics
14:          **end if**
15:          $\mathcal{T} \leftarrow \mathcal{T} \cup \{\psi\}$          $\triangleright$ Add tree to queue
16:          $\mathcal{S} \leftarrow \mathcal{S} \cup \{\psi_{\text{norm}}(X)\}$          $\triangleright$ Add semantics to queue
17:          $\mathcal{H}_{\text{seen}} \leftarrow \mathcal{H}_{\text{seen}} \cup \{\psi_{\text{norm}}(X)\}$          $\triangleright$ Update hash set of seen semantics
18:      **end for**
19: **end for**
20: **if** $|\mathcal{T}| > T_{\text{max}}$ **then**          $\triangleright$ Retain only the most recent $T_{\text{max}}$ entries
21:      $\mathcal{T} \leftarrow \mathcal{T}[-T_{\text{max}} :]$
22:      $\mathcal{S} \leftarrow \mathcal{S}[-T_{\text{max}} :]$
23: **end if**

---

**Algorithm 3** Exact Retrieval

---

1: **Input:** Desired semantics $R$, current tree $\phi_i$, semantic library $\mathcal{L}$, maximum retrieval count $\kappa = 10$
2: **Output:** Proposed tree $\phi_{\text{new}}$
3: $\mathcal{L}_\kappa \leftarrow \text{Top-}\kappa\big(\mathcal{L}, \text{distance}(\mathcal{L}, R)\big)$      $\triangleright$ Retrieve top $\kappa$ trees with smallest distances to $R$
4: $\mathcal{L}_\kappa \leftarrow \text{Sort}(\mathcal{L}_\kappa, \text{distance}(\mathcal{L}_\kappa, R))$      $\triangleright$ Sort by increasing distance to $R$
5: $\phi_{\text{new}} \leftarrow \text{None}$      $\triangleright$ Initialize proposed tree
6: **for** each tree $\phi \in \mathcal{L}_\kappa$ **do**
7:      **if** $\text{size}(\phi) \leq \text{size}(\phi_i)$ **then**
8:          $\phi_{\text{new}} \leftarrow \phi$      $\triangleright$ Select the first valid tree
9:          **Break**      $\triangleright$ Stop further iteration
10:      **end if**
11: **end for**
12: **if** $\phi_{\text{new}} = \text{None}$ **then**
13:      **Return Skip**      $\triangleright$ Skip generation if no valid tree is found
14: **else**
15:      **Return** $\phi_{\text{new}}$      $\triangleright$ Return the proposed tree
16: **end if**

---

are then ranked based on their distance to the desired semantics $R$, and the first tree with the same or fewer nodes than the current tree $\phi_i$ is selected as the proposed tree $\phi_{\text{new}}$. This size constraint encourages the generation of simpler, more interpretable symbolic expressions. If no such tree is found, the current round of retrieval-augmented generation is skipped.

## D    EXACT REPLACEMENT

Exact replacement is a mechanism that replaces features in a solution with more effective features stored in the retrieval library. Since both the features $\phi$ and their associated semantics $\phi(X)$ are stored in the retrieval library $\mathcal{L}$, it is straightforward to determine the optimal scaling factor during

---

**Algorithm 4** Exact Replacement

---

1: **Input:** New proposed tree $\phi_{\text{new}}$, current semantics $\Phi(X)$, proposed updated semantics $\phi_{\text{new}}(X)$, temporary semantics $\Phi(X)^{\text{temp}}$, residual $\mathbf{R}$, target values $\mathbf{y}$
2: **Output:** Updated tree $\phi_i$, updated semantics $\Phi(X)$
3: **if** $\phi_{\text{new}} = \emptyset$ **then**
4:     **continue**                                   ▷ No suitable replacement found
5: **end if**
6: **if** $\frac{\Phi(X)}{\|\Phi(X)\|_2} = \frac{\phi_{\text{new}}(X)}{\|\phi_{\text{new}}(X)\|_2}$ **then**
7:     **continue**                                ▷ Identical semantics; skip replacement
8: **end if**
9: $\beta \leftarrow \frac{(\phi_{\text{new}}(X) - \bar{\phi}_{\text{new}}(X)) \cdot (\mathbf{R} - \bar{\mathbf{R}})}{(\phi_{\text{new}}(X) - \bar{\phi}_{\text{new}}(X))^2}$            ▷ Compute scaling factor
10: $\alpha \leftarrow \bar{\mathbf{R}} - \beta \cdot \bar{\phi}_{\text{new}}(X)$                 ▷ Compute bias term
11: $\Phi(X)^{\text{trial}} \leftarrow \Phi(X)^{\text{temp}} + \beta \cdot \phi_{\text{new}}(X) + \alpha$    ▷ Propose updated semantics
12: $\text{MSE}_{\text{trial}} \leftarrow \frac{1}{|X|} \sum_{j \in X} (\Phi(X_j)^{\text{trial}} - \mathbf{y}_j)^2$
13: $\text{MSE}_{\text{current}} \leftarrow \frac{1}{|X|} \sum_{j \in X} (\Phi(X_j) - \mathbf{y}_j)^2$
14: **if** $\text{MSE}_{\text{trial}} \leq \text{MSE}_{\text{current}}$ **then**
15:     $\phi_i \leftarrow \phi_{\text{new}}$                                  ▷ Accept replacement
16:     $\Phi(X) \leftarrow \Phi(X)^{\text{trial}}$
17: **end if**

---

the replacement process and calculate the new residual after the replacement. The detailed logic is presented in Algorithm 4. Specifically, Lines 9 and 10 describe how to compute the scaling coefficient $\beta$ and bias term $\alpha$, respectively, based on the semantics of the newly proposed feature, $\phi_{\text{new}}(X)$, and the desired semantics, $\mathbf{R}$. The algorithm then calculates and compares the error before and after the replacement, denoted by $\text{MSE}_{\text{current}}$ and $\text{MSE}_{\text{trial}}$, respectively. As shown in Line 15, if the error decreases after the replacement, the new feature is accepted, and the semantics are updated. Otherwise, the original feature is retained.

## E   Traversal Algorithm

**Prediction to Symbolic Tree** The output format of the neural network consists of a sequence of $n_F + n_T$ elements, where the first $n_F$ elements represent functions or terminals, and the remaining $n_T$ elements represent terminals. For example, the output could be $[mul, abs, add, x_1, x_2, x_3]$, which corresponds to the mathematical expression $mul(abs(x_1), add(x_2, x_3))$, a pre-order traversal of a symbolic tree. Thus, to decode the output, a conversion from level-order traversal to pre-order traversal is required, as described in Algorithm 5. Since the neural network does not use an explicit end token, some terminal elements may remain unused after decoding and are discarded. As shown in line 5, a tracking variable $\tau$ is used to monitor whether a full symbolic tree has been formed. Once a complete tree is generated, the decoding process stops, and any remaining tokens are discarded. Additionally, during decoding, if a constant placeholder is predicted (line 18), a random constant is sampled from the uniform distribution $\mathcal{U}[-1, 1]$. This delegates the task of constant tuning to the evolutionary algorithm rather than relying on the neural network to predict the constant accurately.

**Symbolic Tree to Training Target** For training data, symbolic expressions such as $mul(abs(x_1), add(x_2, x_3))$ are represented as $[mul, abs, x_1, add, x_2, x_3]$. However, to serve as proper training labels, this representation must be reordered to reflect a level-order traversal. The algorithm for converting a list of pre-order traversals into level-order representations is detailed in Algorithm 6. The key idea is to collect the required number of child nodes at each level (line 9) and then recursively process these child nodes to obtain subtrees in pre-order traversal (line 13). Finally, the subtrees are assembled into an integral tree in pre-order traversal (line 17).

## F   Neural Network Training Details

### F.1   Parameter Settings

To encourage the neural network to escape local optima, we employ cosine annealing with warm restarts (Loshchilov & Hutter, 2017). Additionally, the Adam optimizer with decoupled weight

---

**Algorithm 5** Level Order Traversal to Preorder Traversal

---
1: **procedure** LEVELORDERTRAVERSAL($\mathcal{N}$)      $\triangleright$ $\mathcal{N}$ is the list of node names
2:      $\tau \leftarrow 0$      $\triangleright$ Initialize tracking variable $\tau$
3:      $\mathcal{S} \leftarrow \emptyset$      $\triangleright$ Initialize an empty list $\mathcal{S}$
4:      **for** each $n_i \in \mathcal{N}$ **do**      $\triangleright$ Iterate over each node $n_i$ in $\mathcal{N}$
5:          **if** $\tau = 0$ **then**      $\triangleright$ Exit loop if a complete tree has been formed
6:              **break**
7:          **end if**
8:          **if** $n_i \in \mathcal{F}$ **then**      $\triangleright$ Check if $n_i$ is a function in $\mathcal{F}$
9:              $f \leftarrow \mathcal{F}[n_i]$
10:             $\alpha \leftarrow \text{arity}(f)$      $\triangleright$ Get the arity $\alpha$ of function $f$
11:             $\tau \leftarrow \tau - 1$
12:             $\tau \leftarrow \tau + \alpha$
13:             $\mathcal{S} \leftarrow \mathcal{S} \cup \{f\}$      $\triangleright$ Append $f$ to list $\mathcal{S}$
14:          **else if** $n_i \in \mathcal{T}$ **then**      $\triangleright$ Check if $n_i$ is a terminal in $\mathcal{T}$
15:             $\tau \leftarrow \tau - 1$
16:             $t \leftarrow \mathcal{T}[n_i]$
17:             **if** $t$ is a constant **then**
18:                 $t \leftarrow \mathcal{U}[-1, 1]$
19:             **end if**
20:             $\mathcal{S} \leftarrow \mathcal{S} \cup \{t\}$      $\triangleright$ Append $t$ to list $\mathcal{S}$
21:          **end if**
22:      **end for**
23:      **return** $\mathcal{S}$      $\triangleright$ Return the final list $\mathcal{S}$ representing the tree
24: **end procedure**

---

**Algorithm 6** Preorder Traversal to Level Order Traversal

---
1: **procedure** PREORDERTRAVERSAL($\mathcal{N}$)
2:      **if** $\mathcal{N} = \emptyset$ **then**      $\triangleright$ Return an empty list if no nodes are left
3:          **return** $\emptyset$
4:      **end if**
5:      $n_{\text{root}} \leftarrow \text{pop}(\mathcal{N}, 0)$      $\triangleright$ Get the first node
6:      **if** $n_{\text{root}}$ is a **Primitive then**      $\triangleright$ Check if the root is a function
7:          $\mathcal{A} \leftarrow \emptyset$      $\triangleright$ Initialize argument list
8:          $\mathcal{R} \leftarrow \emptyset$      $\triangleright$ Initialize root argument list
9:          **for** $i \leftarrow 1$ to arity($n_{\text{root}}$) **do**      $\triangleright$ Collect the function's arguments
10:             $n_i \leftarrow \text{pop}(\mathcal{N}, 0)$
11:             $\mathcal{R} \leftarrow \mathcal{R} \cup \{n_i\}$
12:          **end for**
13:          **for** each $n_i \in \mathcal{R}$ **do**      $\triangleright$ Reorder arguments recursively
14:             $\mathcal{N}_i \leftarrow \text{PreorderTraversal}(\{n_i\} \cup \mathcal{N})$
15:             $\mathcal{A} \leftarrow \mathcal{A} \cup \mathcal{N}_i$
16:          **end for**
17:          **return** $\{n_{\text{root}}\} \cup \mathcal{A}$      $\triangleright$ Return the function followed by its subtrees
18:      **else**
19:          **return** $\{n_{\text{root}}\}$      $\triangleright$ Return the terminal as is
20:      **end if**
21: **end procedure**

---

decay (Loshchilov & Hutter, 2019) is used to mitigate overfitting, with a weight decay parameter of $10^{-4}$. For the MLP initialization, we apply Kaiming initialization (He et al., 2015) to ensure appropriate scaling of the weights. The number of training epochs is set to 1000.

Table 2: Functions and Corresponding Expressions

| Primitive | Expression |
|---|---|
| Add | $a + b$ |
| Subtract | $a - b$ |
| Mul | $a \cdot b$ |
| AQ | $\frac{a}{\sqrt{1+b^2}}$ |
| Sqrt | $\sqrt{a}$ |
| Log | $\log(|a| + 1)$ |
| Abs | $|a|$ |
| Square | $a^2$ |
| Max | $\max(a, b)$ |
| Min | $\min(a, b)$ |
| Neg | $-a$ |
| Sin | $\sin(\pi \times a)$ |
| Cos | $\cos(\pi \times a)$ |

### F.2 TRAINING PROCEDURE

The pseudocode for training the neural network is provided in Algorithm 7. First, the data is split into training and validation sets, as shown in Line 2. The data consists of pairs of semantics and GP trees from the library $\mathcal{L}$, excluding GP trees that exceed the limit on the number of function nodes. $\mathbf{X}_{\text{train}}$ represents the semantics of GP trees, and $\mathbf{y}_{\text{train}}$ represents the nodes of GP trees corresponding to the semantics in $\mathbf{X}_{\text{train}}$. Similarly, $\mathbf{X}_{\text{val}}$ and $\mathbf{y}_{\text{val}}$ are the corresponding validation data with analogous meanings. Next, the data is augmented using the data augmentation strategy introduced in Section 3.3. Subsequently, retrieval trees are prepared for each instance in both the training and validation data.

To avoid trivializing the training task as a simple retrieval task, the KD-Tree is initially constructed using only the training set, as shown in Line 4. When preparing the training data in Line 4, the second-nearest tree is selected as the nearest GP tree to form the sets of retrieved trees, $\mathbf{N}_{\text{train}}$, since the nearest tree corresponds to the GP tree itself. For the validation set, as the KD-Tree includes only the training set in Line 5, the nearest GP tree is used as the retrieved tree to form the set $\mathbf{N}_{\text{val}}$. After retrieving the required samples, the KD-Tree is reconstructed using a combination of the training and validation sets, as shown in Line 7.

To save computational resources during training of the neural semantics library, we employ three strategies. First, the neural network is continuously trained during the evolution process. Second, at the beginning of training, as shown in Line 9, the validation loss is checked to determine if the current validation set loss, $L_{\text{val}}^{\text{current}}$, is smaller than the previous validation set loss, $L_{\text{val}}^{\text{prev}}$. If the validation loss does not degrade, training is skipped to save computational resources. Notably, the construction of the KD-Tree precedes this step, ensuring that the retrieval mechanism can always access the latest information without requiring updates to the neural network. This is a key advantage of RAG-based neural generation. Lastly, the neural network tracks the validation loss throughout the training process. If the validation loss does not improve over $\tau$ iterations, the algorithm performs early stopping to conserve computational resources. These three mechanisms collectively ensure that RAG-SR is an efficient SR algorithm.

## G DETAILS OF GENETIC PROGRAMMING PARAMETERS

The crossover and mutation rates for GP are set to 0.9 and 0.1, respectively. The depth limit of GP trees is set to 10. The function set used in GP is listed in Table 2. To prevent division by zero errors, the division operator is replaced with the analytical quotient operator (Ni et al., 2012). Random constants are drawn from a uniform distribution, $\mathcal{U}[-1, 1]$.

---

**Algorithm 7** Training NN

---

1: **Input:** Training data $\mathcal{D}_{\text{train}}$, batch size $B$, epochs $T$, learning rate $\eta$, validation split $\alpha$, patience $\tau$, last validation loss $L_{\text{val}}^{\text{prev}}$
2: $(\mathbf{X}_{\text{train}}, \mathbf{X}_{\text{val}}, \mathbf{y}_{\text{train}}, \mathbf{y}_{\text{val}}) \leftarrow \text{SplitData}(\mathcal{D}_{\text{train}}, \alpha)$
3: $(\mathbf{X}_{\text{train}}, \mathbf{X}_{\text{val}}, \mathbf{y}_{\text{train}}, \mathbf{y}_{\text{val}}) \leftarrow \text{AugmentData}(\mathbf{X}_{\text{train}}, \mathbf{X}_{\text{val}}, \mathbf{y}_{\text{train}}, \mathbf{y}_{\text{val}})$
4: $(\mathbf{N}_{\text{train}}, \mathbf{X}_{\text{train}}, \mathbf{y}_{\text{train}}) \leftarrow \text{RetrieveTrees}(\mathbf{X}_{\text{train}}, \mathbf{y}_{\text{train}})$
5: $\mathcal{K} \leftarrow \text{BuildKDTree}(\mathbf{X}_{\text{train}}, \mathbf{y}_{\text{train}})$          ▷ $\mathcal{K}$ is the KD-Tree
6: $(\mathbf{N}_{\text{val}}, \mathbf{X}_{\text{val}}, \mathbf{y}_{\text{val}}) \leftarrow \text{PrepareValidationSet}(\mathcal{K}, \mathbf{X}_{\text{val}}, \mathbf{y}_{\text{val}})$
7: $\mathcal{K} \leftarrow \text{BuildKDTree}(\mathbf{X}_{\text{train}} \cup \mathbf{X}_{\text{val}}, \mathbf{y}_{\text{train}} \cup \mathbf{y}_{\text{val}})$    ▷ Reconstruct KD-Tree using full data
8: $L_{\text{val}}^{\text{current}} \leftarrow \text{CalculateValidationLoss}(\mathbf{X}_{\text{val}}, \mathbf{y}_{\text{val}}, \mathbf{N}_{\text{val}})$
9: **if** $L_{\text{val}}^{\text{current}} < L_{\text{val}}^{\text{prev}}$ **then**
10:      **return**                ▷ Skip training if validation loss improves
11: **end if**
12: $L_{\text{val}}^{\text{best}} \leftarrow \infty$                ▷ Initialize best validation loss
13: $C_{\text{patience}} \leftarrow 0$                ▷ Initialize patience counter
14: **for** $t = 1$ to $T$ **do**          ▷ Train using augmented data and retrieved data
15:      $\text{TrainBatch}(\mathbf{X}_{\text{train}}, \mathbf{y}_{\text{train}}, \mathbf{N}_{\text{train}})$
16:      $L_{\text{val}} \leftarrow \text{CalculateValidationLoss}(\mathbf{X}_{\text{val}}, \mathbf{y}_{\text{val}}, \mathbf{N}_{\text{val}})$
17:      **if** $L_{\text{val}} < L_{\text{val}}^{\text{best}}$ **then**
18:          $L_{\text{val}}^{\text{best}} \leftarrow L_{\text{val}}$          ▷ Update best validation loss
19:          $C_{\text{patience}} \leftarrow 0$          ▷ Reset patience counter
20:      **else**
21:          $C_{\text{patience}} \leftarrow C_{\text{patience}} + 1$      ▷ Increment patience counter
22:      **end if**
23:      **if** $C_{\text{patience}} > \tau$ **then**
24:          **Break**               ▷ Early stopping
25:      **end if**
26: **end for**
27: **return**

---

## H   EFFECT OF CONTRASTIVE LEARNING

The contrastive learning mechanism is designed to capture relationships between samples within each batch. To evaluate its effectiveness in enhancing representation learning, we present the pairwise correlation between the encodings of MLP in Figure 9. The first two figures show the correlation of the encoded desired semantics $\mathbf{F}_{\text{MLP}}^{\text{mapped}}$ for 16 randomly selected test instances, with and without contrastive learning, respectively. The third figure shows the raw correlation between desired semantics, while the fourth figure presents the absolute correlation, which is a more appropriate metric for feature construction-based symbolic regression techniques. The results indicate that contrastive learning successfully captures two desired semantics with opposite signs that are highly correlated and represent the same symbolic expression. In contrast, without contrastive learning, the neural network lacks this knowledge and may generate significantly different trees when queried with an opposite sign, as it fails to recognize their equivalence. These results highlight the benefit of using contrastive learning to jointly train the intention encoding and retrieval augmentation encoding components.

## I   SENSITIVITY OF THE WEIGHT OF CONTRASTIVE LOSS

In this section, we analyze the sensitivity of the contrastive loss weight $\lambda$ in RAG-NN. Specifically, we evaluate five different $\lambda$ values: 0.01, 0.025, 0.05, 0.1, and 0.2. The impact of varying $\lambda$ on the edit distance of the test set is shown in Figure 10. The results demonstrate that the weight of the contrastive loss significantly influences accuracy. While the contrastive loss can improve performance, the weight $\lambda$ must be carefully tuned using cross-validation. Otherwise, the contrastive loss may fail to achieve optimal performance. Regarding training time, the effect of different $\lambda$ values on runtime is presented in Figure 11. Overall, the runtime remains stable across different $\lambda$ values. Therefore, when tuning $\lambda$, accuracy should be the primary consideration.

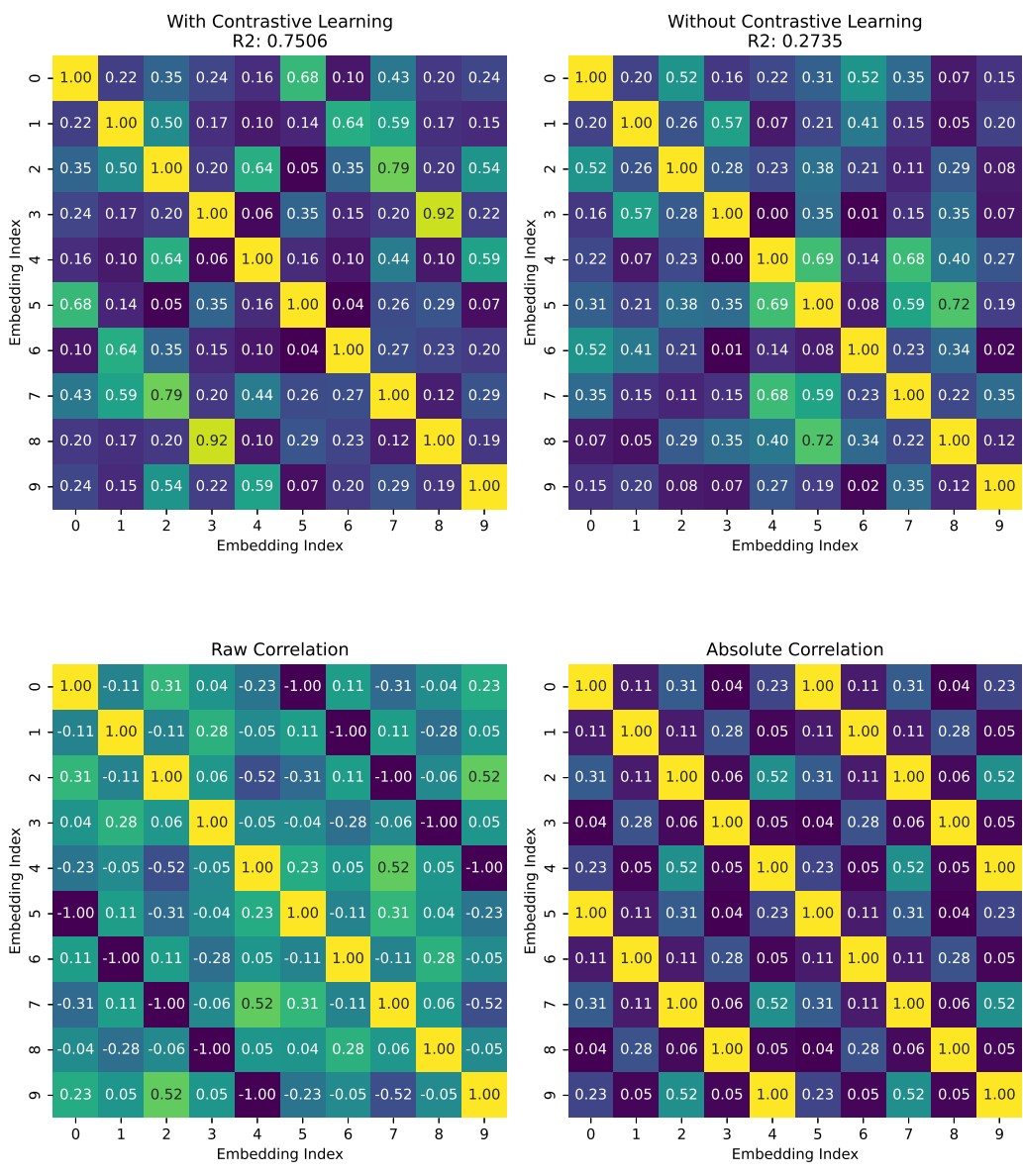

Figure 9: Effect of contrastive learning on pairwise correlation of MLP encodings.

## J  DIFFERENT NUMBER OF VARIABLES

The effect of the proposed components in the neural semantic library may vary with the number of variables. In this paper, we study the impact of the number of variables on the results. The experiments in Section 4.1 are conducted with a variable count of 10. For comparison, additional experiments with variable counts of $\{15, 20, 25, 30, 50\}$ are also performed. Since more variables mean more possible symbolic expressions, the number of symbolic expressions sampled is scaled by the ratio between the number of variables and 10. The results show that the conclusions of the ablation studies in Section 4.1 remain consistent across different numbers of variables. Specifically, both RAG and DA are important contributors to the effectiveness of the neural semantic library. Interestingly, the impact of RAG becomes more significant as the number of variables increases. A possible reason for this trend is that the data might be imbalanced, with some variables being

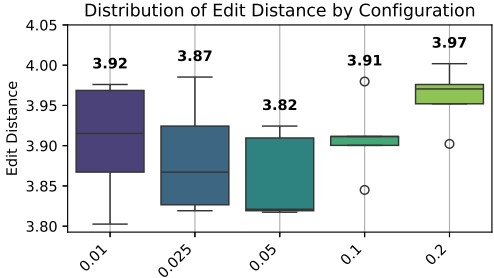
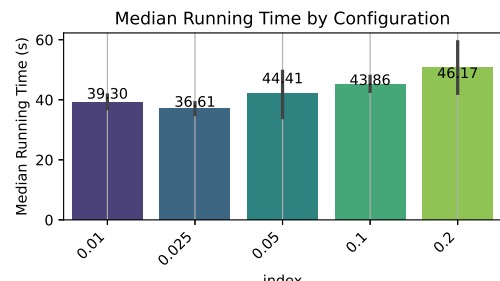

Figure 10: Impact of varying $\lambda$ on edit distance of the test set.

Figure 11: Impact of varying $\lambda$ on running time.

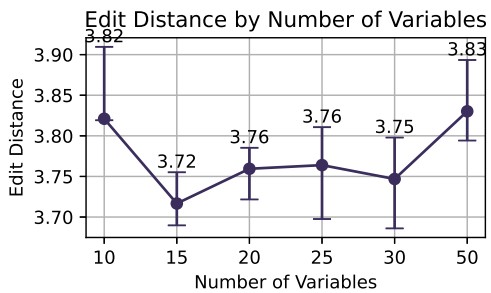
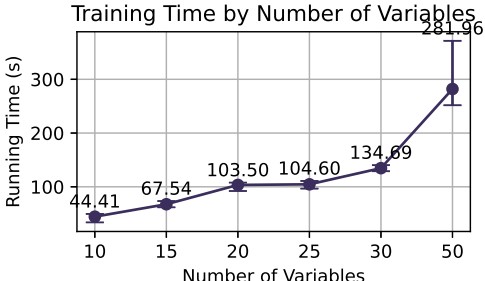

Figure 12: Impact of varying the number of variables on edit distance on the test set.

Figure 13: Impact of varying the number of variables on running time.

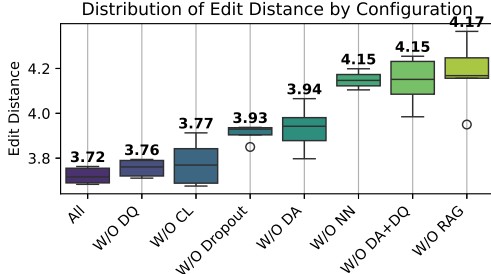
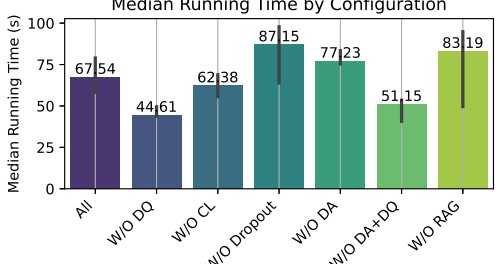

Figure 14: Ablation study of components based on edit distance on the test set (15 variables).

Figure 15: Ablation study of components with respect to running time (15 variables).

underrepresented. In such cases, the retrieval process can extract the relevant variables, even if they represent only a small portion of the data. Therefore, the RAG technique becomes increasingly important as the number of variables increases. More analysis of why RAG could be more effective in high-dimensional scenarios is provided in Appendix K.

To better visualize how the number of variables influences the effectiveness of the neural semantic library, we present the change in edit distance with respect to the number of variables in Figure 12 and the running time with respect to the number of variables in Figure 13. The results indicate that the model can maintain a stable level of prediction accuracy with different number of variables. This indicates that the proposed method demonstrates good scalability for tasks with varying numbers of variables. In comparison, the end-to-end Transformer for symbolic regression (Kamienny et al., 2022) can only be used to train tasks with up to 10 variables. Regarding training time, an increase in the number of variables does lead to a longer training time; however, the increase is modest. These results demonstrate that the neural semantic library is also efficient for tasks with different numbers of variables.

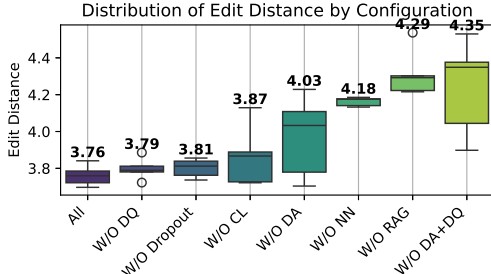

Figure 16: Ablation study of components based on edit distance on the test set (20 variables).

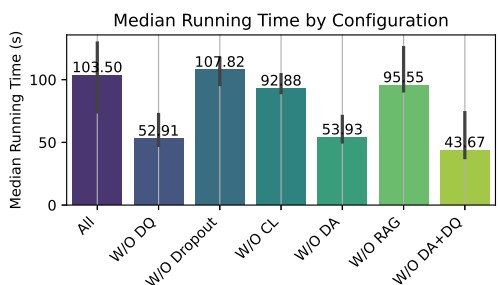

Figure 17: Ablation study of components with respect to running time (20 variables).

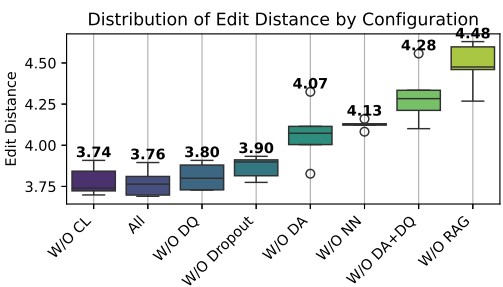

Figure 18: Ablation study of components based on edit distance on the test set (25 variables).

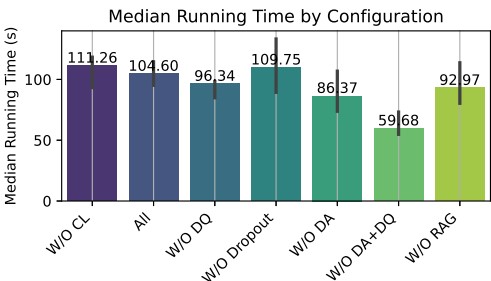

Figure 19: Ablation study of components with respect to running time (25 variables).

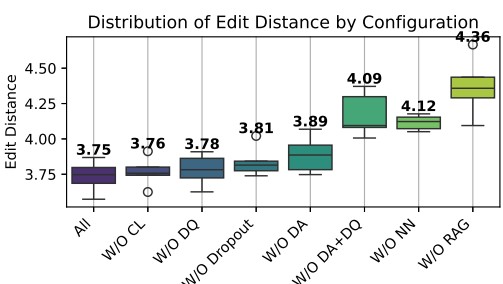

Figure 20: Ablation study of components based on edit distance on the test set (30 variables).

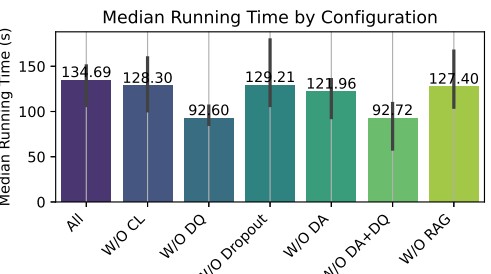

Figure 21: Ablation study of components with respect to running time (30 variables).

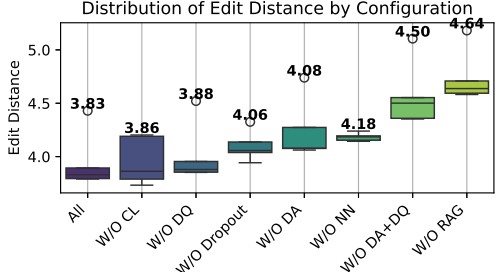

Figure 22: Ablation study of components based on edit distance on the test set (50 variables).

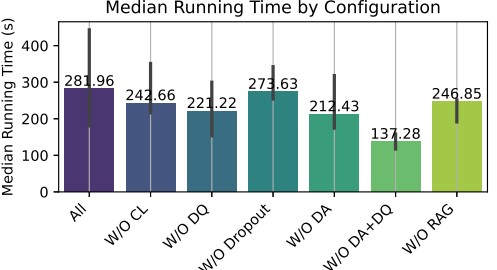

Figure 23: Ablation study of components with respect to running time (50 variables).

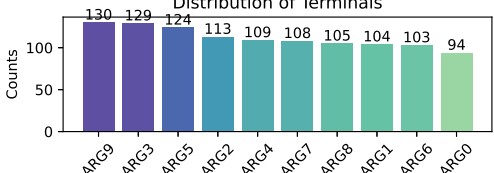

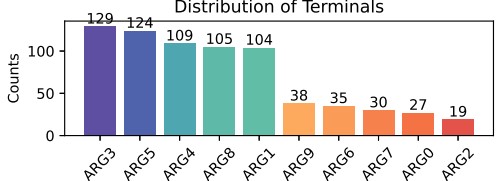

Figure 24: Distribution of terminals with 10 variables.

Figure 25: Distribution of terminals with 10 variables and an imbalance ratio of 0.25.

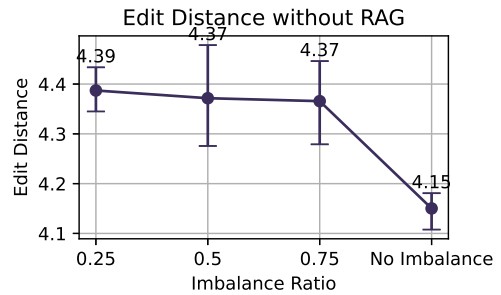

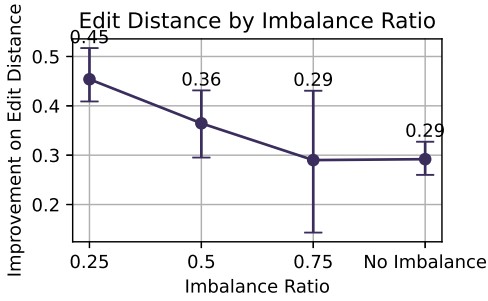

Figure 26: Edit distance across different imbalance ratios without retrieval-augmented generation.

Figure 27: Performance improvement using retrieval-augmented generation across different imbalance ratios.

## K   RETRIEVAL AUGMENTATION FOR IMBALANCED DATA

In Appendix J, RAG has been shown to be particularly beneficial when handling a large number of variables. One potential explanation is the increased data imbalance as the number of variables grows. Specifically, the frequency distribution of variables at position 3 (starting from 0) for 10 and 50 variables is depicted in Figure 24 and Figure 28, respectively. With 10 variables, the most frequent variable appears 49 times and the least frequent appears 37 times, showing a relatively balanced distribution. However, for 50 variables, the most frequent variable appears 26 times while the least frequent variable appears only 7 times, indicating a significant imbalance. This confirms that larger variable sets lead to more imbalanced data distributions.

To validate the effectiveness of the RAG mechanism in handling imbalanced data, we randomly subsample half of the terminals with varying ratios to create imbalanced data for the case of 10 variables. For training data with selected terminals at position 3, subsampling retained the data with a probability equal to the subsampling ratio $s$. The subsampling ratios are $[0.25, 0.5, 0.75]$. Figure 25 shows the distribution after subsampling with a ratio of 0.25. The experimental results are shown in Figure 26 and Figure 27. As shown in Figure 26, the edit distance worsens as the subsampling ratio increases, confirming that imbalanced training data negatively impacts performance. The results in Figure 27 demonstrate that the performance improvement from using RAG becomes more pronounced as the subsampling ratio increases. This indicates that RAG effectively mitigates the issues associated with imbalanced data, explaining why its impact is more significant when dealing with larger numbers of variables.

## L   FURTHER ANALYSIS ON THE SYMBOLIC REGRESSION BENCHMARK

### L.1   COMPARISON ON LOW-DIMENSIONAL BLACK-BOX DATASETS

In the domain of deep symbolic regression, low-dimensional subsets of black-box datasets are widely used as evaluation criteria. Specifically, we select all datasets from the PMLB benchmark with 10 or fewer variables and no categorical features (Kamienny et al., 2022; Shojaee et al., 2024a; Meidani et al., 2024). Figure 29 shows the results under these criteria. Compared to Figure 5, the rel-

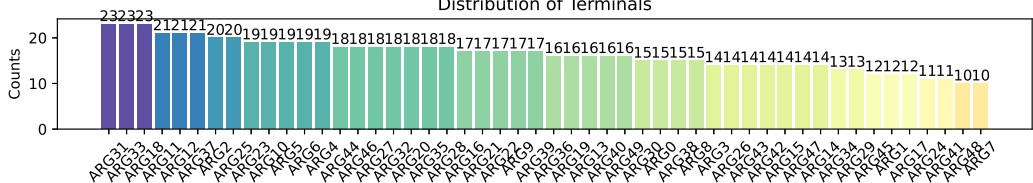

Figure 28: Distribution of functions and terminals in the case of 50 variables.

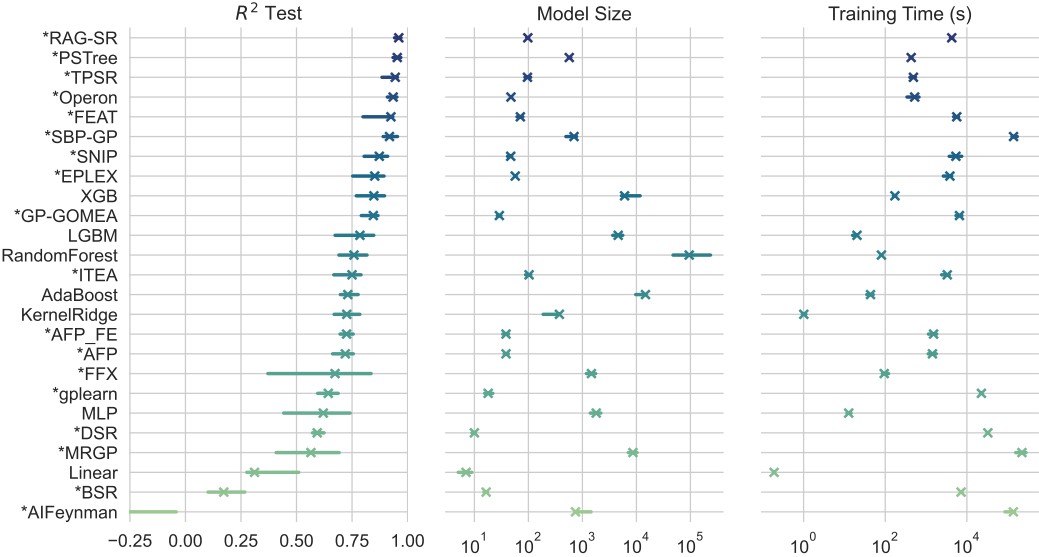

Figure 29: $R^2$ scores, model sizes, and training time of 25 algorithms on low-dimensional black-box regression problems.

ative rankings of the algorithms remain largely unchanged. Therefore, the conclusions in Section 4.2 remain valid in this case.

## L.2 COMPARISON ON FEYNMAN AND STROGATZ DATASETS

**Experimental Settings** The Feynman and Strogatz datasets are widely used synthetic benchmarks for symbolic regression. AI-Feynman consists of 119 datasets, while Strogatz contains 14 datasets. For RAG-SR, we employ a function set relevant to the physics domain, including the operations $+, -, *, /, \sin, \cos$. The division operator is implemented as a protected division to handle edge cases, capping extremely small positive and negative denominators at $10^{-10}$ and $-10^{-10}$, respectively. The training protocol follows the SRBench standard. All baseline algorithms use 10,000 subsampled training instances from the AI-Feynman datasets (La Cava et al., 2021). However, our method subsamples only 500 instances from the Feynman datasets to reduce computational costs. No noise is added to the labels. For the Strogatz datasets, we use all available training instances. Following the definition in SRBench (La Cava et al., 2021), accuracy is defined as 1 if the $R^2$ value exceeds 0.999, and 0 otherwise.

**Experimental Results** The experimental results on the Feynman datasets are shown in Figure 30. Algorithms are ranked based on their test $R^2$ scores on the Strogatz datasets. As shown, RAG-SR achieves the highest test $R^2$ scores on the Strogatz datasets, demonstrating the strong learning capability of the proposed method. On the Feynman datasets, RAG-SR ranks second, trailing only MRGP (Arnaldo et al., 2014). However, the model size of MRGP is more than an order of magnitude larger than RAG-SR, while RAG-SR achieves a good balance. In comparison to deep learning-based symbolic regression methods such as TPSR (Shojaee et al., 2024a) and SNIP (Meidani et al., 2024),

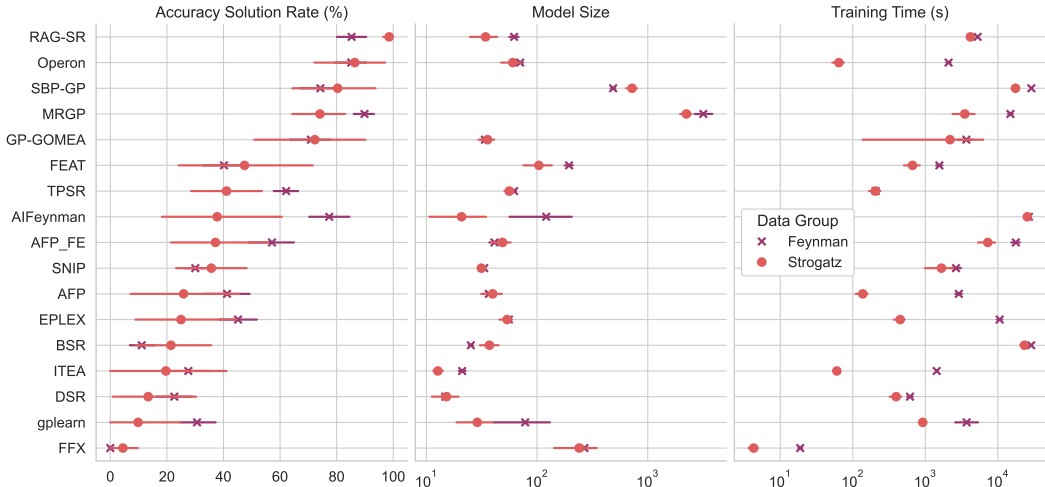

Figure 30: $R^2$ scores, model sizes, and training time of 17 algorithms on 119 Feynman and 14 Strogatz datasets.

RAG-SR outperforms them by a large margin, further validating the effectiveness of leveraging a neural semantic library in symbolic regression.

# M    PARAMETER SENSITIVITY ANALYSIS

## M.1    INFLUENCE OF $P_{\text{NEURAL}}$

$P_{\text{neural}}$ controls the rate at which the neural semantic library is used to generate new individuals. In this section, we analyze the influence of $P_{\text{neural}}$ after evaluating 2000 solutions. Each probability value of $P_{\text{neural}}$ was tested on 120 black-box datasets, with each experiment repeated 30 times to ensure stability. The results, shown in Figure 31, demonstrate that the test $R^2$ scores gradually improve as the probability of using the neural semantic library increases. Since neural generation is more time-consuming than retrieval, the default value of 0.1, which corresponds to applying neural generation to one out of ten GP trees in each solution during replacement, is a reasonable choice that provides a good balance between accuracy and efficiency. Nonetheless, in cases where a GPU is available to provide faster inference, a higher value of $P_{\text{neural}}$ can yield better results.

## M.2    INFLUENCE OF NUMBER OF TREES

The number of trees determines the number of constructed features in each solution. RAG-SR follows conventions in the evolutionary feature construction literature, using 10 trees as the default parameter. In this section, we compare three configurations for the number of trees in each solution: 5, 10, and 15. To ensure stability, each configuration was tested on 120 black-box datasets, with each experiment repeated 30 times. The results after evaluating 2000 solutions are shown in Figure 32. The results indicate a substantial performance gap between using 10 trees and using 5 trees. Several factors may contribute to this improvement. First, a larger number of GP trees in each solution provides more candidates to be stored in the library, allowing neural generation and exact retrieval to identify better GP trees. Additionally, having more trees in each solution enables the base learner to more flexibly select appropriate features for making predictions, which may further enhance performance. However, increasing the number of trees also leads to an increase in model size and evaluation time for each solution, as shown in Figure 33 and Figure 34. Therefore, setting the number of trees to 10 strikes a good balance between accuracy, model size, and training time.

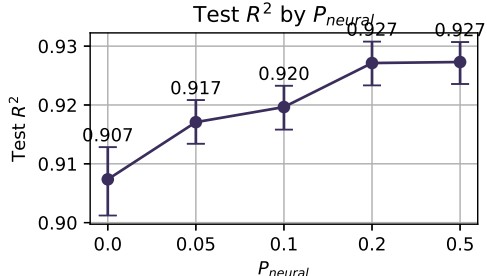

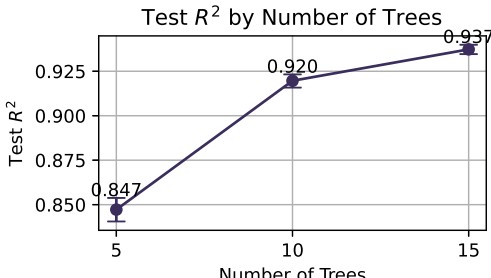

Figure 31: Sensitivity of test $R^2$ score to $P_{\text{neural}}$.

Figure 32: Sensitivity of test $R^2$ score to number of trees.

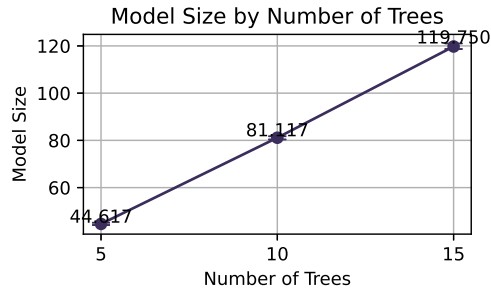

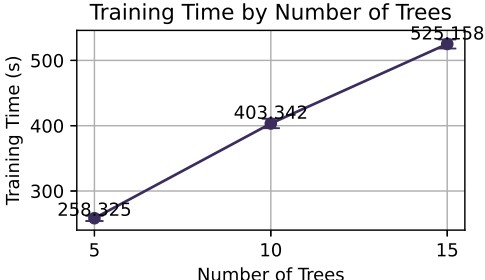

Figure 33: Sensitivity of model size to number of trees.

Figure 34: Sensitivity of training time to number of trees.

## N    RELATED WORK

### N.1    RAG FOR CODE GENERATION

The Retrieval-Augmented Generation (RAG) technique has been widely used in the domain of code generation to enhance accuracy by retrieving code from open-source databases (Parvez et al., 2021; Wang et al., 2024). Similarly, RAG has also been applied to code transformation tasks, demonstrating improved performance compared to zero-shot code translation (Bhattarai et al., 2024). These RAG-based techniques typically rely on existing repositories (Parvez et al., 2021). In this paper, we demonstrate that RAG is not restricted to pre-existing code repositories but can also be applied to code generated by evolutionary algorithms. Furthermore, we show that data augmentation can improve prediction accuracy when using RAG in conjunction with neural networks for symbolic expression generation. This suggests that, for code generation, employing synonym replacement or paraphrasing models to augment code summaries could further enhance the efficiency of RAG-based systems.

### N.2    VQ-VAE FOR CODE GENERATION

In the program synthesis domain, Vector Quantized-Variational Autoencoder (VQ-VAE) (Van Den Oord et al., 2017) has been employed to learn discrete representations from input-output pairs, enabling the capture of high-level concepts (Hong et al., 2021). Both VQ-VAE and RAG leverage continuous semantics to retrieve discrete information for code generation. However, VQ-VAE utilizes an autoencoder to learn high-level conceptual representations from data, whereas RAG directly retrieves relevant information from a database. Compared to VQ-VAE, the RAG technique offers greater interpretability for end-users by allowing inspection of the retrieved information. Moreover, RAG provides an efficient mechanism to update knowledge without requiring retraining of the neural network. Although there are many differences between VQ-VAE and RAG, these techniques are not mutually exclusive. Integrating them in the context of neural symbolic regression presents a promising avenue for future research.

Table 3: Examples of symbolic trees generated by the retrieval-augmented neural network when the ground truth exceeds the generation limit.

| RAG-NN Generated Tree (Distance) | Ground Truth Tree |
|---|---|
| Log(Log(Log(ARG1))) (3) | Abs(Log(Neg(Abs(Log(Log(ARG1)))))) |
| Cos(Sin(Cos(ARG5))) (3) | Log(Abs(Cos(Sin(Sqrt(Cos(ARG5)))))) |
| Log(Abs(Log(ARG7))) (3) | Log(Abs(Sqrt(Abs(Sqrt(Log(ARG7)))))) |
| Sin(Square(Sin(ARG8))) (3) | Neg(Cos(Abs(Sin(Square(Sin(ARG8)))))) |
| Log(Cos(ARG8)) (4) | Cos(Log(Sin(Cos(Log(Cos(ARG8)))))) |
| Sqrt(Sqrt(ARG4)) (4) | Sin(Log(Log(Sqrt(Log(Sqrt(ARG4)))))) |
| Log(Sin(ARG1)) (4) | Log(Neg(Cos(Sqrt(Sqrt(Sin(ARG1)))))) |
| Log(Cos(Log(ARG5))) (4) | Cos(Log(Neg(Sqrt(Cos(Neg(ARG5)))))) |
| Log(Square(Log(ARG5))) (4) | Log(Square(Log(Neg(Log(AQ(ARG7, ARG5)))))) |
| Sqrt(Sqrt(ARG2)) (4) | Sqrt(Neg(Abs(Sqrt(Sqrt(Square(ARG2)))))) |

## N.3 LARGE LANGUAGE MODELS FOR SYMBOLIC REGRESSION

Recently, large language models (LLMs) have been increasingly applied to symbolic regression tasks. These approaches often involve leveraging historical programs (Shojaee et al., 2024b) or extracting patterns from historical programs (Grayeli et al., 2024) to guide LLM-based symbolic regression through prompt engineering. By instructing LLMs to learn from high-quality historical programs (Shojaee et al., 2024b) while avoiding low-quality ones (Grayeli et al., 2024), these methods aim to leverage LLMs to implement effective SR systems. However, using the MSE score as the criterion for crafting prompts can be misleading. Some expressions with higher MSE scores may still serve as valuable building blocks. From the perspective of the semantic space, the most useful expression for a candidate solution is not necessarily the one with the lowest MSE but the one that effectively fills the residual gap. This insight highlights the need for a fine-grained retrieval augmentation strategy that focuses on the semantics of each promising solution when crafting prompts. Thus, this paper builds a library of GP trees on semantics and retrieves from this library to guide the language model in generating useful expressions for each promising solution.

## O OUT-OF-LIMIT GENERATION

The neural retrieval library is trained on symbolic expressions with heights ranging between $[0, 5]$. However, in an SR task, the ground truth expression may exceed the generation limit of the neural model. In this section, we evaluate the performance of the neural retrieval library when the ground truth surpasses this limit. Specifically, we generate GP trees with a height of 6 and compare the outputs of RAG-NN with the ground truth, as shown in Table 3. The results indicate that even when the ground truth exceeds the neural generation limit, the neural model can still generate meaningful sub-expressions that assist GP in discovering better GP trees.

