# OpenReview forum: "RAG-SR: Retrieval-Augmented Generation for Neural Symbolic Regression"
_ICLR.cc/2025/Conference — ICLR 2025 Spotlight_

### Official Review · Reviewer_b3Mt · 2024-11-03

**Soundness:** 3
**Presentation:** 3
**Contribution:** 3
**Rating:** 8
**Confidence:** 3

**Summary:**

This paper introduces a novel approach to neural-guided symbolic regression. While prior work has focused on training generative models to directly produce new expression trees, such models often suffer from hallucination and struggle when faced with out-of-distribution data. In contrast, the authors propose training the generative model to produce a single feature at a time, rather than generating an entire expression tree. This approach aims to improve the model's performance on out-of-distribution problems. Additionally, the authors incorporate a prompting mechanism, feeding the model with relevant expression trees to help reduce hallucination.

The authors evaluate their method (RAG-SR) on a custom synthetic dataset and the no-noise version of SRBench. They find that RAG-SR outperforms other baselines on these task in terms of accuracy, size of the final equation, and speed.

**Strengths:**

- The general strategy of using a set of discrete latent codes for robustness is sound and widely acknowledged as beneficial in practical applications (e.g., VQ-VAE, discrete World Models, etc.).
- The paper is well written with great figures.

**Weaknesses:**

- **Clarity**: Many related ideas exist within the broader community, and while they may not necessarily be weaknesses, these papers should be discussed in the related work section:

  - VQ-VAE (https://arxiv.org/pdf/1711.00937): This paper from the representation learning community learns a discrete latent representation (a _codebook_). The authors might benefit from reviewing this work. I believe the manuscrpt is more closely tied to "learning a discrete representation for neural guided symbolic regression" rather than "retrieval augmented generation." Regardless, I highly recommend adopting the nomenclature in this paper to increase comprehensiveness. This technique has also been applied in synthesizing general-purpose programs (https://arxiv.org/pdf/2012.00377), a comparision with this work would be useful.

  -  Since this work is heavily influenced by work in retrieval augmented generation, it would be useful to include a discussion on how the authors think their work is connected/influenced by work in RAG (https://arxiv.org/pdf/2406.14497) and how it's different from current work that uses RAG for code generation (https://arxiv.org/pdf/2108.11601, https://arxiv.org/pdf/2407.19619).


Regardless, I believe the authors have adequately demonstrated that this is novel and performant algorithm for symbolic regression. As such, I am in favor of accepting this paper.

**Questions:**

- L71: "pretraining free supervised learning:" I'm a bit confused by this phrase. Isn't pretraining, as used colloquially in NLP, a self-supervised learning strategy? Under this definition, all supervised learning algorithms should always be "pretraining free." Consider replacing this phrase.
- L103: Considering this paper has "RAG" in the title, it might be useful to have a section which introduces RAG, how it's relevant to symbolic regression / program synthesis problems, and how it's relevant to this algorithm. I suggest making a general section on "Foundation Models for program induction." These papers (the RAG paper, RAG + code generation papers, and symbolic regression + foundation models) might be useful:
  - https://arxiv.org/pdf/2406.14497: RAG Paper
  - https://arxiv.org/pdf/2108.11601, https://arxiv.org/pdf/2407.19619 : RAG + Code generation
  - https://arxiv.org/abs/2404.18400, https://arxiv.org/abs/2409.09359 : RAG + Foundation Models
- L198: Is the depth and number of trees fixed? How does the algorithm scale when $m$ is increased/decreased? How is $m$ selected for the experiments (Synthetic benchmark and SRBench)?
- L195-L209: "SD focuses on replacing existing trees in the model with more informative ones..."  Is the residual a tensor of real numbers or a single real number? Also, how is the difference measured? Regardless, it seems that calculating the residual requires evaluating the performance of the equation without the subterm to be replaced. For many domains where SR is used, evaluating the performance of a program is very expensive (eg: computational weather models).
- L206: What is the definition of the semantics $\psi(X)$? Is it a vector (e.g., $\psi(X) \in \mathbb{R}^N$), and is it learned, or does it represent subtree performance on indices in $X$?
- L210: How is P_neural computed? This isn't mentioned in the algorithm or in the following section. Is it a hyper-parameter?
- L409: Please add error bars to the runtime evaluation experiments by averaging across different seeds.
- L1177: Many SRBench ground-truth equations have depths greater than five. What do the failure modes on this experiment look like? What do the discovered equations look like? How do they compare to the ground truth equations (edit distance)?

---

> ### Author Response · Authors · 2024-11-22
>
> **Q1.**
>
> Thank you for your suggestion. To improve clarity, we have replaced "pretraining-free supervised learning" with "supervised learning methods that do not rely on pretraining" to emphasize that RAG-SR does not require pretraining.
>
> **Q2./W.**
>
> Thank you for suggesting these relevant papers. We have included discussions on VQ-VAE for code generation, RAG for code generation, and LLMs for symbolic regression in Section N of the supplementary material, highlighting their relevance and differences from our work. Regarding the differences between VQ-VAE and RAG-NN, there are two key distinctions:
>
> - **Interpretable Retrieval Results:**  RAG-SR retrieves symbolic expressions from a library, offering human-interpretable solutions, whereas VQ-VAE learns high-level concepts through a discrete latent representation.
>
> - **Access to Latest Information Without Retraining:**  RAG-SR can utilize the latest information even without training the neural network on new data. Specifically, new symbolic expressions can be added to the retrieval library to provide up-to-date information. The neural network does not need to be retrained as long as validation performance does not degrade.
>
> While RAG and VQ-VAE are different, they can be complementary and combined to enhance code generation from multiple perspectives, which could be explored as a potential direction for future research.
>
> **Q3.**
>
> The maximum depth of GP trees and the number of trees are both fixed at 10, consistent with common settings in the GP literature. The algorithm maintains these fixed numbers throughout the evolutionary process.
>
> To investigate scalability, we have conducted an analysis of the impact of the number of GP trees, included in Section M.2 of the supplementary material. Results indicate that setting the number of trees to 10 strikes a good balance between accuracy, model size, and training time, making it a suitable default choice.
>
> **Q4.**
>
> The residual is a vector of real numbers. To minimize computational overhead, we pre-compute and store the semantics of all subtrees for each solution during the evaluation phase. This allows us to avoid re-evaluating the semantics of the GP trees during the replacement process.
>
> Specifically, if the semantics of a solution is represented by  $\hat{Y} = \beta_1 \phi_1(X) + \dots + \beta_m \phi_m(X)$, the updated semantics after removing tree $k$ can be computed as $\hat{Y} - \beta_k \phi_k(X)$, and the residual is given by $Y - (\hat{Y} - \beta_k \phi_k(X))$. When replacing a tree, the process uses pre-computed semantics $\phi_1(X), \dots, \phi_m(X)$, ensuring efficiency. For a tree $\sigma$ retrieved from the semantics library, the new error, $[Y - (\hat{Y} - \beta_k \phi_k(X)+\beta_\sigma \sigma(X))]^2$, can be efficiently calculated using the stored semantics of $\sigma(X)$ from the library. For a tree $\sigma$ generated by the neural network, we assume the neural network generates good trees and skip error evaluation to reduce computational overhead.

---

> ### Author Response · Authors · 2024-11-22
>
> **Q5.**
>
> Semantics are defined as the output of applying a function $\phi$ to the input data $X$. For a GP tree $\phi$, the semantics are represented as $\phi(X)$, denoting the result of executing the GP tree on $X$. For a solution $\Phi$, semantics are defined as the output of the entire model, which includes both constructed features and the linear model, formally given by $\Phi(X) = \beta_1 \phi_1(X) + \dots + \beta_m \phi_m(X)$. In both cases, semantics are represented as a vector where each element corresponds to the output for a specific training instance.
>
> The semantics determine the solution's performance based on a metric such as mean squared error. We have revised the paper to make it clear that semantics refer to the output of a model $\Phi$, a tree $\phi$, or a subtree $\psi$ on the input data $X$.
>
> **Q6.**
>
> The probability $P_{\text{neural}}$, which controls the balance between generating new expressions using the neural network and retrieving symbolic expressions from the semantic library, is a hyperparameter.
>
> In this work, $P_{\text{neural}}$ is set to 0.1, guided by the rationale that one tree in each solution is replaced by a neural network-generated tree, as there are ten trees in each solution. An analysis of the influence of $P_{\text{neural}}$ has been added to Section M.1 of the supplementary material. Empirical results confirm that $P_{\text{neural}} = 0.1$ achieves an effective balance between neural generation and retrieval, ensuring effectiveness while minimizing the computational burden of neural generation.
>
> **Q7.**
>
> We have added error bars to all figures related to runtime results. After analyzing these error bars, we confirm that our conclusions regarding runtime performance remain unchanged.
>
> **Q8.**
>
> Although neural networks are limited to training on a small number of function nodes, the height of GP trees is capped at 10 to allow for more complex ground-truth equations. In this scenario, the role of the neural network is to provide useful building blocks, allowing the evolutionary algorithm to refine them into the final expressions.
>
> To investigate what the neural network generates when the ground truth exceeds the limits of neural generation (i.e., the failure modes), we have added an analysis in Section O of the supplementary material. Experimental results for neural generation in cases where the ground-truth tree height is 6 have been included, including the generated expressions and the edit distance to the ground truth. These results show that, even when the ground truth exceeds the limits of neural generation, the neural network contributes valuable subexpressions that aid the evolutionary search process.

---

> > ### Comment · Reviewer_b3Mt · 2024-11-22
> >
> > Thank you for the response. I'm maintaining my current score. Great work!

---

### Official Review · Reviewer_e4Uw · 2024-11-04

**Soundness:** 3
**Presentation:** 3
**Contribution:** 3
**Rating:** 8
**Confidence:** 4

**Summary:**

The paper proposes RAG-SR, a retrieval-augmented generation framework for symbolic regression (SR) that utilizes a neural semantic library, semantic descent, and scale-invariant data augmentation to generate symbolic models effectively without pre-training. Through various experiments, the paper demonstrates that RAG-SR outperforms existing SR models, especially those relying on pre-trained language models, by producing interpretable and accurate symbolic expressions.

**Strengths:**

1. **Innovative Retrieval-Augmented Generation (RAG)**: The RAG mechanism effectively addresses the common issue of model hallucinations in SR tasks by incorporating relevant expressions from the semantic library. This approach is well-justified and experimentally validated, showing a significant positive impact on both the quality and relevance of generated models.

2. **Effective Feature Construction through Semantic Descent**: The proposed semantic descent strategy is a notable contribution that improves model refinement through selective feature replacement rather than addition, which helps maintain a compact and interpretable feature set while boosting accuracy.

3. **Data Augmentation and Double Query Strategy**: These techniques address the inherent challenges in SR by ensuring scale-invariance and robustness, with the double query strategy, in particular, effectively capturing symmetrical relationships in symbolic expressions.

4. **Strong Empirical Validation**: The paper demonstrates that RAG-SR achieves state-of-the-art results across diverse benchmarks, reinforcing the proposed method's efficacy. The study includes a well-structured comparison with relevant baselines, supporting claims of improved performance and interpretability.

**Weaknesses:**

1. **Complexity and Training Time Trade-offs**: While RAG-SR is shown to improve SR accuracy, the additional computational overhead from maintaining and querying the semantic library may be a concern, particularly for larger datasets or real-time applications. It would be beneficial to discuss potential optimizations or alternative approaches to mitigate the added computational cost.

2. **Limited Generalization to Non-SR Domains**: The focus on SR limits the method’s applicability to other machine learning tasks. A discussion on the adaptability of RAG-SR to different types of regression or generative tasks, particularly those that do not rely on symbolic expressions, would add value to the paper's impact on broader ML applications.

3. **Dependence on Hyperparameter Tuning**: The effectiveness of RAG-SR, particularly in terms of balancing the retrieval and neural generation probabilities, appears sensitive to hyperparameter choices. The paper could benefit from including a sensitivity analysis or guidelines on setting these parameters in diverse settings.

4. **Interpretability of Generated Models**: Although RAG-SR produces interpretable symbolic models, the paper could improve by further quantifying interpretability. For instance, how do the generated models compare in complexity or comprehensibility against baselines from a domain expert’s perspective?

**Questions:**

1. **Explain Hyperparameter Choices and Sensitivity**: Could the authors provide more insights into the choice of hyperparameters, such as the probability $\( P_{\text{neural}} \)$ for neural generation versus retrieval, and the value of $\( \lambda \)$ for the contrastive loss term? A sensitivity analysis on these hyperparameters would be valuable for practical deployment.

2. **Computational Cost Analysis**: The results indicate increased computational costs with RAG-SR. Could the authors discuss how the computational demands scale with larger datasets? Are there specific optimizations for the semantic library or model architecture that could reduce inference times?

3. **Ablation on Data Augmentation and Double Query**: Data augmentation and double querying are innovative and appear to significantly contribute to the model's effectiveness. However, would these techniques still benefit RAG-SR in less structured or non-SR tasks? Clarifying this point would highlight the broader applicability of the approach.

4. **Comparison with Pre-trained Language Models**: While RAG-SR surpasses pre-trained models in SR tasks, does the method show robustness against varied functional distributions, especially in cases with substantial function overlap? A breakdown of results based on function novelty (seen vs. unseen functions) would deepen the insights into RAG-SR’s advantages.

5. **Interpretable Complexity Metrics**: Interpretability is a key focus in SR, yet model complexity is only briefly addressed. How does RAG-SR ensure a balance between interpretability and accuracy, especially as model complexity scales? Including complexity metrics or visualizations would substantiate claims on interpretability.

**Details Of Ethics Concerns:**

Not at all

---

> ### Author Response · Authors · 2024-11-22
>
> **Q1./W3.**
>
> Thank you for your comments. We have included a hyperparameter sensitivity analysis in Sections I and M of the supplementary material in the revised version.
>
> - **Analysis on the Probability of Neural Generation $P_{\text{neural}}$:**  In this work, $P_{\text{neural}}$ is set to 0.1, guided by the rationale that one tree in each solution is replaced by a neural network-generated tree, as there are ten trees in each solution. An analysis of the influence of $P_{\text{neural}}$ has been added to Section M.1 of the supplementary material. Empirical results confirm that $P_{\text{neural}} = 0.1$ is an effective choice, providing a good trade-off between neural generation and retrieval from the knowledge base. This setting ensures effective use of neural generation while minimizing its computational burden.
>
> - **Analysis on the Weight of the Contrastive Loss $\lambda$:**  As for the weight of the contrastive loss $\lambda$, five different values ranging from 0.01 to 0.2 are evaluated to study their effect on accuracy and runtime, as detailed in Section I of the supplementary material. The results indicate that runtime is not significantly affected by $\lambda$, while accuracy is the primary factor that should be considered when tuning $\lambda$.
>
> **Q2./W1.**
>
> To improve scalability on large datasets, RAG-SR employs instance selection every 10 generations, retaining the semantics of the top 50 most challenging instances based on loss rankings. This ensures that the time consumption for the retrieval and neural generation components of RAG-SR remains consistent regardless of dataset size, making the algorithm highly scalable. Specifically, this strategy provides two key benefits for large datasets:
>
> - **Reducing Query Complexity:**  This approach reduces the query complexity of the KD-Tree during retrieval, with a worst-case time complexity of $O\left(k \cdot n^{1 - \frac{1}{k}}\right)$ [2] for $k$-dimensional semantics (i.e., $k$ instances).
>
> - **Reducing Training Complexity of the Neural Network:**  Fixing the dimensionality of semantics via instance selection ensures that the neural network input size remains constant, thereby facilitating scalability to large datasets.
>
> Further details are included in Section A of the supplementary material.
>
> **Q3./W2.**
>
> Data augmentation and double querying are not limited to SR tasks. For example, in RAG-based code generation, text queries can be augmented by paraphrasing using appropriate tools. A broader discussion on this applicability has been added to Section N.1 of the supplementary material.
>
> **Q4.**
>
> We agree that evaluating the robustness of language model-based SR methods against varied functional distributions and function overlap is important. In this work, RAG-SR has not been pre-trained on external datasets. Consequently, both the black-box experiments and experiments on the Feynman and Strogatz datasets represent unseen problems for RAG-SR, demonstrating its ability to generalize to novel tasks.
>
> **Q5./W4.**
>
> In RAG-SR, interpretability is measured by the number of nodes in the symbolic tree, a commonly used metric for estimating model complexity in domains such as physics [1]. RAG-SR incorporates two mechanisms to balance interpretability and accuracy:
>
> - **Complexity Control in Exact Retrieval:**  First, during exact retrieval and replacement, the top-10 nearest trees are retrieved, and the nearest tree that is smaller than or equal in size to the incumbent GP tree is selected for replacement, rather than directly using the nearest tree.
>
> - **Complexity Control in Neural Generation:**  Second, the neural network is constrained to generate trees within a limited size.
>
> These measures ensure a good balance between simplicity and accuracy.
>
> [1]. Tenachi, W., Ibata, R., \& Diakogiannis, F. I. (2023). Deep symbolic regression for physics guided by units constraints: toward the automated discovery of physical laws. The Astrophysical Journal, 959(2), 99.
>
> [2]. Lee, D. T., \& Wong, C. K. (1977). Worst-case analysis for region and partial region searches in multidimensional binary search trees and balanced quad trees. Acta Informatica, 9(1), 23-29.

---

> ### Comment · Reviewer_e4Uw · 2024-11-25
> **Thank you for the rebuttal**
>
> I have read the rebuttal and willing to keep my score.

---

### Official Review · Reviewer_aRJ2 · 2024-11-04

**Soundness:** 3
**Presentation:** 2
**Contribution:** 3
**Rating:** 6
**Confidence:** 3

**Summary:**

The paper proposes a novel approach to symbolic regression based on neural-network based Retrieval Augmented Generation (RAG) and semantic descent (SD). The main idea is to dynamically construct a semantic library associating trees with their corresponding semantics (numerical evaluations of the tree). Semantic Descent starts from a set of initial trees and calculates the residual, i.e. the numerical error obtained from removing a specific tree from the original tree set. The goal of this step is to understand the contribution of the missing tree and replacing it with a new tree, possibly improving the residual. The proposal of a new tree is based on querying the semantic library with the residual and (optionally) refining such a tree with the prediction of a neural network conditioned on both the proposed tree from the semantic library and an appropriate encoding of the residual. A simple MLP is responsible for obtaining the latter encoding. A transformer encoder processes the tree extracted from the library and produces another embedding. The embeddings from the MLP and the transformer encoder are then concatenated and fed into a transformer decoder which in turn outputs the final proposed tree. The whole model is trained with online supervised learning. The library is dynamically updated with first-in-first-out queue mechanism. A data augmentation strategy is also proposed to improve the robustness of the model. Experiments are divided into two parts: first the model is evaluated on synthetic data and the various components of the model are ablated to quantify their relative importance and their contribution to the final running time; second the model is tested on the PLMB benchmark, resulting in very good results both in terms of accuracy and complexity.

**Strengths:**

- SR-RAG represents, to best of the reviewer knowledge, a novel method for symbolic regression combining techniques from the genetic programming literature and recent advances in neural networks. This combination is interesting as the two components have the potential to nicely address each other's limitations. In particular, the ablation studies seem to show that neural generation performs better than simple retrieval, suggesting that the neural network is benefitting the whole approach.

- Results are encouraging: the model performs very well in terms of R^2 score and complexity.

**Weaknesses:**

- **Clarity of presentation could be improved**: I believe that the way the approach is presented should be made clearer. In particular, Fig. 1 is a bit hard to interpret and does not aid the reader understanding. In addition, I feel more details should be given on how the semantic library is populated and how the retrieval is performed given a residual.

- **I feel the way the neural network is trained is not explained in sufficient detail**: While the way the different neural network components are described is well done, I feel important details on their training are missing. In particular, I do not understand how the cross-entropy loss is used as it is not clear to me what the ground truth labels are and where they are taken from. In addition, more details about the online supervised learning training routine should be reported. I feel this is very important to precisely understand how to model is trained and to have an intuition of the overall complexity of the approach. I would appreciate if the authors could clarify these points.

- **Question on runtime**: RAG-SR performs relatively well in terms of training time, even though it incorporates neural networks and models are trained on CPU. I am wondering why possibly less complex models, exclusively based on GP techniques, perform comparably or worse in terms of runtime, despite not incorporating any neural network.

- **Question on complexity**: RAG-SR seems to perform well in terms of complexity, achieving a good trade-off between accuracy and complexity. I am wondering where this property comes from. Why does the model has a preference toward simpler expression. At which point of the model is this constraint imposed?

**Questions:**

See weaknesses section.

---

> ### Author Response · Authors · 2024-11-22
>
> **Q1.**
>
> Thank you for your feedback. Figure 1 provides an overview of the key components of our approach, illustrating the interactions among them and emphasizing the role of the neural network in generating a new tree that is highly correlated with the residual. To improve clarity, we have reorganized the flow of Figure 1 and expanded it with more descriptive labels and details.
>
> - **Semantic Library Update Procedure:**  We have expanded the explanation in Section B of the supplementary material to describe the details of how the semantic library is populated, including the update procedure, the criteria for storing new entries/trees, and the data structure used for efficient retrieval. Specifically, the library is updated at the end of each generation during the evolutionary process to maintain a set of GP tree semantics (stored in a queue $S$) and their corresponding GP trees (stored in a queue $T$), using a First-In-First-Out queue mechanism to store the most recent entries. A hash set ($H_{\text{seen}}$) tracks previously seen semantics to prevent duplicate entries.
>
> - **Retrieval Process in Exact Replacement and Neural Generation:**  To provide further clarity on the retrieval process, we have included pseudocode and additional explanations in Section C of the supplementary material. These describe how residuals are matched to trees in the semantic library and how the most appropriate tree is identified in exact replacement. Specifically, the retrieval algorithm retrieves the top $ \kappa = 10 $ symbolic trees with the smallest distances to the residual $R$ in the retrieval library. If a tree that is smaller than or equal to the incumbent GP tree is found, it will be selected as the proposed tree ($\phi_{\text{new}}$). Otherwise, the current round of retrieval will be skipped.
>
>     For the retrieval process in neural generation, it relies on a KD-Tree built based on all symbolic expressions with a maximum of $n_T$ function nodes. The KD-Tree enables efficient querying of the nearest semantics given the desired semantics $ R $. The index of the nearest semantics will then be used to retrieve the corresponding symbolic expression from the library.
>
> **Q2.**
>
> For additional clarity regarding the training of the neural network, we have provided the following updates and clarifications to address these concerns.
>
> - **Use of Cross-Entropy Loss:**  We have provided an explanation in Section E of the supplementary material to clarify the definition of the ground truth labels used in the cross-entropy loss. Specifically, the ground truth labels include functions such as $+$, $*$, and terminals like $x_1$, $x_2$. A list of these functions and terminals corresponds to the GP tree $\phi$ that achieves the semantics $\phi(X)$ and is taken from the retrieval library. These ground truth labels are encoded as sequences in a pre-order traversal format of symbolic trees. For instance, a sequence like [mul, abs, add, $x_1$, $x_2$, $x_3$] corresponds to the mathematical expression mul(abs($x_1$), add($x_2$, $x_3$)). During training, the neural network predicts sequences similar to the pre-order traversal of the symbolic tree. The cross-entropy loss compares the predicted sequence with the ground truth sequence, token by token. For each position in the sequence, the predicted probability distribution over all possible tokens (output by the neural network) is compared to the one-hot encoding of the ground truth token for that position.
>
> - **Training Procedure of RAG-NN:**  For the neural network training procedure, a detailed explanation and pseudocode are provided in Section F of the supplementary material for further clarification. We explain how the neural network continuously updates its weights during the evolutionary process. Specifically, the following processes are performed sequentially:
>
>     - **Data Preparation:**  First, based on pairs of GP trees and semantics from the semantic library, the data preparation stage includes splitting the data into training and validation sets, augmenting the data, and using a KD-Tree for efficient retrieval of symbolic trees.
>
>     - **Semantic Library Update:**  Next, before training the neural network, a validation check is performed. If the validation loss does not degrade, only the retrieval library is updated to allow the neural network to access the latest information without further training.
>
>     - **Neural Network Update:**  Finally, if the validation loss degrades, additional training is performed. The neural network is trained iteratively using the Adam optimizer with cosine annealing and early stopping based on validation loss.
>
> This procedure ensures that the neural network adapts dynamically to changes in the evolutionary process and efficiently learns from the evolving data.

---

> ### Author Response · Authors · 2024-11-22
>
> **Q3.**
>
> - **High Efficiency in Retrieval:**  Regarding the runtime performance of RAG-SR compared to GP-only models, the baseline algorithm, SBP-GP [1], while not incorporating neural networks, can still exhibit high runtime because it enumerates a large set of GP trees in an external library to determine the best GP trees for replacement, resulting in a time complexity of $O(L)$, where $L$ is the number of GP trees in the library. In contrast, RAG-SR uses a KD-Tree for efficient retrieval, reducing the time complexity to $O(\log L)$.
>
> - **High Efficiency in Neural Network Training:**  Additionally, RAG-SR improves efficiency by monitoring performance on a validation set, skipping neural network training during certain generations, incrementally training the neural network, and employing early stopping. The KD-Tree is reconstructed at each generation, irrespective of whether training is skipped. Thanks to the RAG mechanism, the neural network accesses information from the latest trees in the evolution process via retrieval from the library, even without further training. These optimizations collectively enable RAG-SR to achieve better runtime efficiency.
>
> **Q4.**
>
> Regarding the balance between accuracy and complexity achieved by RAG-SR, it arises from two size constraint mechanisms in our framework:
>
> - **Complexity Control in Exact Retrieval:**  First, during exact retrieval and replacement, as outlined in Algorithm 3, the KD-Tree selects trees based on their distance to the residual, and a size constraint ensures that only trees equal to or smaller than the current tree are considered, promoting simplicity.
>
> - **Complexity Control in Neural Generation:**  Second, the neural network is constrained to generate trees within a predefined size limit, further contributing to concise expressions.
>
>
> [1]. Virgolin, M., Alderliesten, T., \& Bosman, P. A. (2019, July). Linear scaling with and within semantic backpropagation-based genetic programming for symbolic regression. In Proceedings of the Genetic and Evolutionary Computation Conference (pp. 1084-1092).

---

### Author Response · Authors · 2024-11-22

We sincerely thank all reviewers for dedicating your time and providing constructive comments. We have uploaded the revised manuscript and highlighted all changes in blue. We hope these revisions address your concerns and look forward to your feedback.

---

### Meta-Review · Area_Chair_9sEC · 2024-12-17

**Metareview:**

This paper focuses on symbolic regression and proposes an iterative algorithm which uses a neural network to replace symbolic expressions from a library at every step. This replacement is done by either proposing a new expression from scratch or retrieving an existing old one (this is the RAG aspect). The experiments show performance improvements over baselines and ablations show that the RAG component is helpful in mitigating hallucinations.

**Additional Comments On Reviewer Discussion:**

The reviewers didn’t have major concerns with the paper and the consensus is to accept it.

---

### Decision · Program_Chairs · 2025-01-22

Accept (Spotlight)